# Maximising environmental savings from silicon photovoltaics manufacturing to 2035

Bethany L. Willis [1], Oliver M. Rigby [1], Sophie L. Pain [2], Nicholas E. Grant[2], John D. Murphy [2,3], Ruy S. Bonilla [4] & Neil S. Beattie [1]✉

The silicon photovoltaics market is transitioning from the incumbent passivated emitter rear cell to the higher efficiency tunnel oxide passivated contact technology and it is crucial to understand the environmental impact of this change. Here, we conduct life cycle assessment to compare both technologies quantitatively and identify environmental savings in 15 of 16 environmental impact categories for tunnel oxide passivated contact. This includes a 6.5% reduction in carbon dioxide equivalent emissions, per watt peak at the expense of 15.2% increase in metal resource use, for photovoltaic modules manufactured in China and transported to central Europe. A critical factor in photovoltaics manufacturing is the carbon intensity of the electricity mix. We model the impact of photovoltaics production across different global regions, incorporating future electricity mix scenarios and a projection for photovoltaics deployment. Our model provides a forecast of the environmental impact of global photovoltaics manufacturing and identifies a potential reduction of 8.2 gigatonnes of carbon dioxide equivalent emissions by 2035, depending on manufacturing location.

Rapid, global decarbonisation requires swift transitions towards higher-performing, sustainable, renewable energy technologies. Photovoltaics (PV) offers enormous promise as a low-carbon[1], versatile[2] and relatively inexpensive[1,3] technology. Recently, PV has been deployed at an unprecedented scale, reaching over 1 terawatt peak (TW$_p$) of cumulative installed capacity by the end of 2023[4]. This growth is expected to continue at scale such that total installed capacity could reach 80 TW$_p$ by 2050[5,6]. However, this growth requires additional resources and causes environmental impact. Previous work has quantified resources required for terawatt-scale PV production to 2100[7], finding that material demand could exceed capacity depending on silicon architecture, though this doesn't consider the environmental impact of material extraction. Other work seeks to quantify material demand whilst considering electricity demand and greenhouse gas emissions[5], finding up to 11% of the 1.5 °C greenhouse gas emission budget could be spent on terawatt PV production. Whilst much existing PV research focuses on greenhouse gas emissions, it neglects broader environmental impacts such as ecosystem health, other atmospheric impacts, and human health, which need to be considered to truly evaluate the environmental sustainability. This type of analysis can be achieved using Life Cycle Assessment (LCA).

LCA has been applied to PV to identify areas of high environmental impact ("hotspots")[8], investigating factors such as silicon feedstock[9], wafer size or module design[10,11]. LCA has also been used to compare different technologies[12] and evaluate supply chains[13]. Fewer studies compare silicon cell architectures, with the majority focusing on general comparisons between mono-crystalline and multi-crystalline (which is no longer a timely issue) technologies, usually without stating cell architecture. Although comparisons of aluminium back surface field to passivated emitter rear cell (PERC) technology exist[14], these are quickly becoming outdated due to rapid technological advancements. Our work considers the environmental impact of currently leading silicon architectures, which will dominate the PV market over the next decade.

[1]School of Engineering, Physics and Mathematics, Northumbria University, Newcastle upon Tyne, UK. [2]School of Engineering, University of Warwick, Coventry, UK. [3]School of Engineering, University of Birmingham, Edgbaston, Birmingham, UK. [4]Department of Materials, University of Oxford, Oxford, UK. ✉e-mail: neil.beattie@northumbria.ac.uk

Recently, mainstream silicon PV technology has shifted from PERC to tunnel oxide passivated contact (TOPCon) cells. Manufacturing TOPCon cells is similar to PERC, though there are significant structural differences resulting in higher efficiency. Firstly, PERC tends to use a gallium-doped p-type wafer, whereas TOPCon uses an n-type dopant, such as phosphorus or antimony. At the front, PERC uses a phosphorus diffused emitter and silicon nitride ($SiN_x$) as a combined passivation layer and anti-reflection coating, whilst TOPCon uses a boron-doped emitter with an aluminium oxide ($AlO_x$) passivation layer and $SiN_x$ antireflective coating. At the rear, PERC uses $AlO_x$ to passivate the p-type surface of the wafer before being capped with $SiN_x$, whereas TOPCon uses a tunnel oxide layer, improving passivation, coated by a phosphorus-doped poly-silicon layer and capped with $SiN_x$, improving contact conductivity[15]. Finally, the contacts differ: silver is used for the front in both PERC and TOPCon, but the former uses a combination of silver and aluminium for the rear contact whereas the latter uses just silver.

Unlike PERC, the environmental sustainability of TOPCon modules is relatively underexplored in the literature. The only other assessment of TOPCon manufacturing using LCA focuses on manufacturing solely in China[16]. The work investigated environmental differences between mono and bi-facial modules, p-type and n-type technologies, wafer size and carbon emissions from PV manufacturing between 2023 and 2060. Although the work provides a detailed comparison between module designs within China, TOPCon manufacturing outside China remains unexplored.

Here, using LCA, we explore the sustainability of TOPCon manufacturing, suitably addressing the literature gap, by identifying hotspots regarding future PV manufacturing and performing a quantitative comparison to PERC. The impact is considered on a global scale by uniquely including projections for technology and materials consumption improvements from the International Technology Roadmap for Photovoltaics (ITRPV), as well as electricity mix scenarios from the US Energy Information Administration, into our LCA. Additionally, we provide a readily accessible and industrially validated life-cycle inventory for TOPCon cell manufacturing as a basis for other researchers to build upon and develop further modelling. A cradle-to-gate approach is adopted, considering raw material extraction, module assembly, and transportation to central Europe. Conducting Monte Carlo uncertainty analysis provides additional confidence in our conclusions. We evaluate the impact of TOPCon manufacturing up to 2035 whilst considering technological developments and future electricity mix scenarios for different manufacturing locations: India, China, the United States of America (US) and Europe. Sensitivity of TOPCon manufacturing to the electricity mix composition is investigated, demonstrating its critical relevance. Subsequently, we forecast the cumulative impact of PV deployment over the coming decade and compare its benefits to the environmental cost of manufacturing using LCA. The Supplementary Information contains assumptions (Supplementary Table 1) and raw data, while the Supplementary Data contains life cycle inventories and calculations used throughout the work and all figure data can be accessed in the Source Data.

## Results

### State-of-the-art module manufacturing

A manufactured PV module comprises a wafer, cell, module components and transportation of the module to the deployment location (in this case, central Europe). The TOPCon module life-cycle inventory is created by adapting a PERC inventory (Supplementary Data 1 and 2)[11] to account for different doping and include primary cell data from an international TOPCon manufacturer. The contribution of each production stage is investigated for 16 environmental impact categories, described in a glossary (Supplementary Table 2).

Here, a baseline scenario compares a 1 $W_p$ PERC and TOPCon module manufactured in China and transported to central Europe,

considering the technological state in 2023. LCA results (Supplementary Table 3) show TOPCon modules exhibit lower impacts than PERC for 15 of 16 investigated environmental impact categories (per $W_p$). The exception is *Resource use (minerals and metals)*, which is 15.2% higher for TOPCon, due to increased silver used for TOPCon contacts, compared to the silver and aluminium mix adopted in PERC. Results are normalised to the annual impact of an average European, using the Environmental Footprint v3.1 methodology, to identify the six highest value impact categories and simplify the presentation of results. Normalisation is only done for the "PERC vs TOPCon" comparison and hotspot analysis. After normalisation, the six highest value impact categories are identified as: *Climate change, Particulate matter, Eutrophication (freshwater), Photochemical ozone formation, Resource use (fossils)* and *Resource use (minerals and metals)* as shown in Fig. 1, together with the relative percentage change between PERC and TOPCon modules. For simplicity, *Resource use (fossils)* and *Resource use (minerals and metals)* will be referred to as *"Fossil use"* and *"Metal use"*, respectively. All 16 categories are normalised and presented in Supplementary Fig. 1. A key cause for improved environmental impact is the reduced amount of material per $W_p$ of TOPCon due to higher cell conversion efficiency.

Figure 2 presents a breakdown of the TOPCon module manufacturing stages, shown in Fig. 1, for the hotspot analysis. For TOPCon, the wafer dominates 12 of 16 environmental impact categories. These categories have large contributions from the electricity consumed during silicon purification, providing >85% of the wafer impact in 10 categories. A breakdown of the wafer impact for the six highest impact categories in Fig. 2a shows that electricity consumption dominates all six. High impact categories in wafer production correlate with increased impact for overall module production: *Fossil use, Climate change* and *Particulate matter*, where electricity consumption during wafer production accounts for 88.2%, 89.9% and 93.5%, respectively. This highlights electricity consumption during wafer production as a hotspot, particularly the production of poly-silicon (poly-Si) and Czochralski-silicon (Cz-Si). These high impacts stem from the use of fossil-fuel sources for electricity generation highlighting the importance of increasing the share of renewables in the electricity mix to reduce these impacts. However, the use of renewable energy still contributes towards various other environmental impacts, although not as large as fossil fuels, so efforts should also target minimising electricity consumption during silicon purification.

*Metal use* has the highest value and is the only category dominated by cell fabrication, as shown in Fig. 1. The TOPCon cell fabrication impact is broken down in Fig. 2b. High *Metal use* values are from silver use during metallisation of (front and rear) contacts, representing 53.0% of the whole module *Metal use* impact, and 98.3% of the cell *Metal use* impact. The magnitude of *Metal use* is four times larger than the next highest impact category, *Photochemical ozone formation*. This sets a quantitative target for research and development activities to reduce silver consumption and adoption of alternative contact materials, such as copper[17]. Emissions and waste treatment during cell production make up 83.4% of the *Photochemical ozone formation* category and 48.6% of *Particulate matter*, notably non-methane volatile organic compounds (NMVOC) emissions and particulates <2.5 μm. The cause of these emissions is not explicit since inventory items for direct emissions have been taken from the literature[11] though NMVOC emissions are typically associated with using solvents and cleaning processes[18], suggesting that optimising or minimising solvent use would reduce these emissions. Other hotspots include poly-Si deposition (specifically silane usage) and annealing (specifically electricity consumption), which contribute >15% of the impact in 11 impact categories. Silane also dominates silicon nitride deposition during plasma-enhanced chemical vapour deposition (PECVD) at the cell front and rear. Rear PECVD contributes >15% to two impact categories, whereas the Front PECVD impact is greater, contributing >15% to six

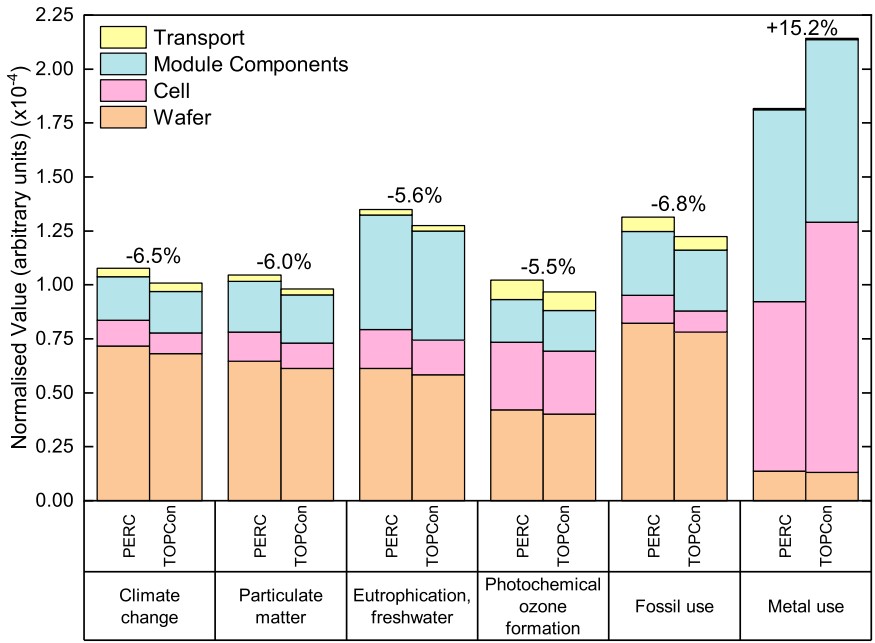

**Fig. 1 | Normalised environmental impact of manufacturing 1 $W_p$ PERC and TOPCon modules.** Normalised results showing the six highest impact categories associated with the manufacturing of 1 $W_p$ PERC and TOPCon module and the relative percentage change. This considers manufacturing in China using an average electricity grid mix and transportation to central Europe.

impact categories due to trimethylaluminum usage. Recovering unused silane[19] or minimising its use[20] through reducing poly-Si thickness are routes to reducing this impact. However, changes to the device structure, such as poly-Si thinning, may adversely affect the cell performance. This trade-off should be carefully considered by performing further LCA that considers these structural changes and consequent performance to ensure synergistic impact reduction and performance improvement.

The module components' production is the largest contributor to two impact categories: *Human toxicity (non-cancer)*, and *Land use* (see Supplementary Fig. 1). A breakdown of the module components' impact in Fig. 2c identifies hotspot materials as copper and solar glass, in addition to tin, which impacts *Metal use* due to its scarcity. Copper is a hotspot for multiple categories, providing >50% of the overall module impact in three categories. Solar glass also contributes highly, representing >25% of the module components' impact in eight categories and >50% in two further categories. Upstream solar glass production analysis identifies uncoated flat glass as high impact, specifically using soda ash, electricity and heavy fuel oil during production. Reducing the associated impact through recycling key materials used during PV production, such as copper, glass, silver, and silicon, has been investigated previously[21,22].

Module transportation from China to Europe requires freight lorries, trains and ships. LCA results in Fig. 2d show that the freight ship dominates all six high-impact categories, though it transports modules over 100 times the distance transported by lorry and train. Comparing 1 tkm (tonne-kilometre) of each transportation method (see Supplementary Table 4) demonstrates the smaller impact of transoceanic freight relative to other forms of transportation. Transport has the largest impact on photochemical *ozone formation*, due to hydrocarbon fuels used for shipping and lorry transport and construction of rail infrastructure. Additionally, the *Fossil use* category has a large impact resulting from the use of fossil fuels (e.g., petroleum, diesel, hard coal and heavy fuel oil) to power the vehicles.

## Technology development and manufacturing location

The environmental impact of manufacturing PERC is sensitive to the module performance and manufacturing location[11]. For the following results, we assume that the processing yield and manufacturing capacity of PV in China can be replicated in the other investigated locations. We investigate the impact of 1 $W_p$ TOPCon manufacturing for India, China, the US and Europe. Different locations are considered by changing the electricity mix, such that it represents the composition of each location and modifying transport inputs accordingly, from the manufacturing location to central Europe. We consider *low zero-carbon technologies cost scenario*[23] from the Energy Information Administration (EIA) to represent decarbonisation of each location's future electricity mix. Additionally, the results encompass effects of performance and material advancements on TOPCon manufacturing to 2035 – according to ITRPV 2024[24], these values are provided in Supplementary Table 5. The impact of manufacturing TOPCon (per $W_p$) reduces over time due to these improvements as shown in Fig. 3 for the six highest impact categories.

In 2023, TOPCon manufacturing in India has the highest Climate change impact (0.95 kg $CO_2$ equivalent (eq.) $W_p^{-1}$), whilst the lowest occurs in Europe (0.40 kg $CO_2$ eq. $W_p^{-1}$), shown in Fig. 3a. By 2035, assuming electricity mixes remain constant (solid lines), *Climate change* decreases on average by 0.10 kg $CO_2$ eq. $W_p^{-1}$ across all locations due to improved performance, reduced silver during metallisation and reduced poly-Si for Cz-Si production. This reduction for *Climate change* is more sensitive to poly-Si reduction (and related electricity reduction) than silver reduction, which instead greatly effects *Metal use*.

Considering all scenarios in 2023–2034, TOPCon manufacturing can result in 0.31–0.95 kg $CO_2$ eq. $W_p^{-1}$, with the lowest impact in Europe. Analysis of dashed lines shows the impact of TOPCon manufacturing reducing further when considering future electricity mix scenarios which account for progress towards decarbonisation.

The five other high-impact categories (Fig. 3b–3f) show similar trends to *Climate change*, with India and Europe tending to have the highest and lowest impact, respectively. *Particulate matter* (Fig. 3b) differs, where China's impact is greater than India (for current and future scenarios) despite India having a larger share of coal in its mix. This is due to high quantities of particulate emissions released during electricity production for internal use during coal mining in China. Also, the US has the highest *Metal use* impact (considering future

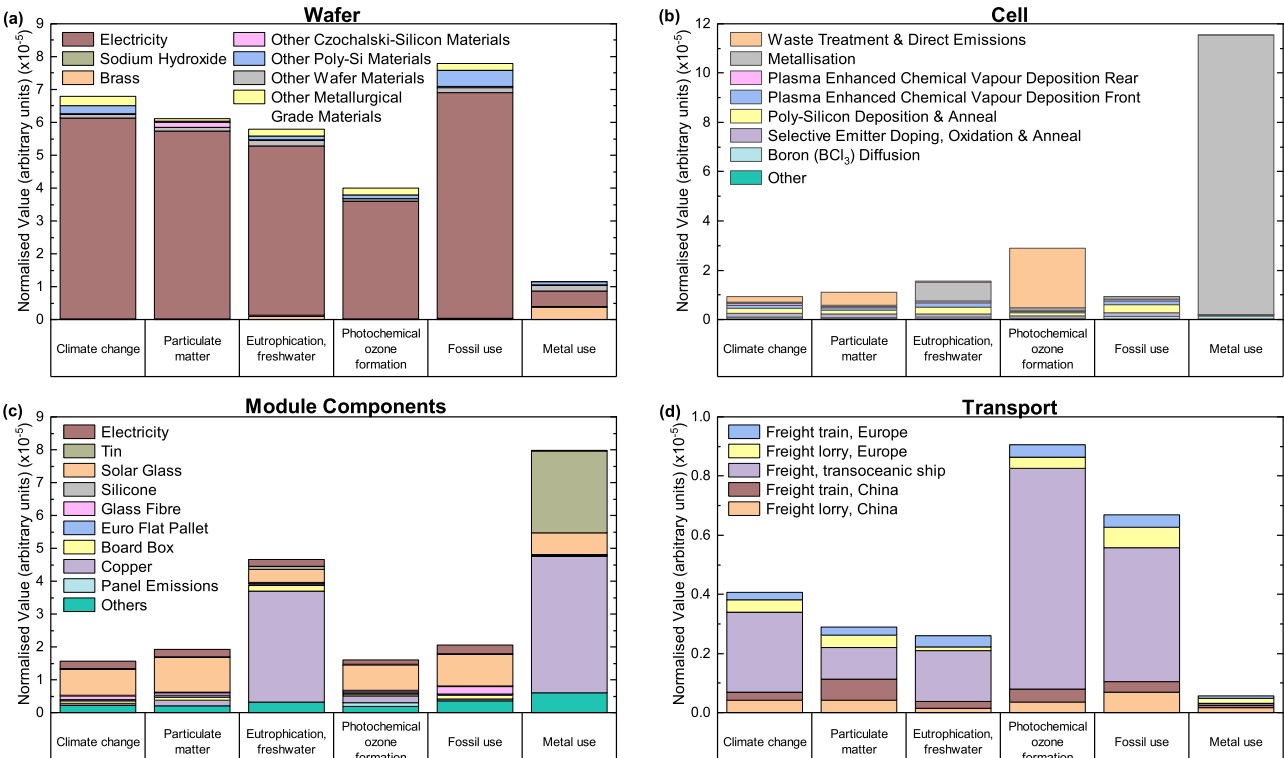

**Fig. 2 | Normalised environmental impact of 1 $W_p$ TOPCon module manufacturing stages.** Normalised results showing the individual material and process contributions towards the six identified high-impact categories for **a** TOPCon wafer production, **b** TOPCon solar cell fabrication, **c** TOPCon module component manufacturing and **d** transportation from China to central Europe. Due to substantial inventories, "Others" represents materials contributing to all impact categories <15% in (**a**) and <5% in (**b**) and (**c**). This considers manufacturing in China using an average electricity grid mix and transportation to central Europe.

electricity mixes) by 2035 (Fig. 3f). This is a consequence of an increased share of renewable resources, where more scarce materials, such as silver, are required for renewable energy systems. Scarce materials have higher *Metal use* values than abundant materials. This increased proportion of renewable energy decreases the proportion of fossil-based resources, causing a complementary reduction in US *Fossil use*, shown by the blue dashed line in Fig. 3e.

When considering future electricity mixes (dashed lines), all impact categories in Fig. 3 exhibit greater reductions in TOPCon manufacturing, except *Metal use* (Fig. 3f), which slightly increases due to an increasing share of renewables. Europe has the lowest *Metal use* impact for the future mix because the proportion of PV in the mix is lowest relative to other locations. There is still a significant overall reduction in *Metal use* by 2035 due to decreasing silver consumption, emphasising its criticality for reducing this impact.

It should be noted that current electricity mix models use the Ecoinvent database, where India's inventory is over five years old, whilst other countries were updated at the end of 2023. To account for this discrepancy, future scenarios also include a 2023 datapoint. China shows a higher *Eutrophication (freshwater)* impact for the future electricity mix than the current mix, though both scenarios show agreement that the impact will decrease over time. This suggests that there are differences between modelled electricity mixes such that electricity sources that contribute more to *Eutrophication (freshwater)* are greater in the future scenario than the current (Ecoinvent) scenario. This could be from the use of hard coal in the mix, which was found to significantly contribute to this category.

**Global deployment**
Confirming that the benefits of global PV deployment outweigh the environmental costs of manufacturing is essential. The environmental cost of manufacturing is modelled based on PERC and TOPCon

deployment to 2035, taken from ITRPV 2024[24], for *Climate change* and *Metal use* impacts shown in Fig. 4. The shaded region acts as an indicator of the difference in impact between current and future electricity mixes. The benefits of this global PV deployment are described in the following section, where $CO_2$ emissions associated with each kWh are evaluated.

*Climate change* (Fig. 4a) shows that PV deployment to 2035 can result in up to 13.8 Gt $CO_2$ eq., but changes to manufacturing location and decarbonising, future electricity mixes can reduce this by 8.2 Gt $CO_2$ eq. For context, this saving is equivalent to 13.9% of global net anthropogenic greenhouse gas emissions in 2019[25]. Additionally, our results suggest that changing manufacturing location from China to Europe could reduce this impact by 49.5% to 2035, assuming EIA scenarios hold true. However, the renewable technology deployment rate in various locations, particularly China[26], presents a high level of uncertainty which should be addressed through collective development of more detailed and accessible electricity mix scenarios.

*Metal use* (Fig. 4b) increases with the decarbonising nature of future electricity mixes for all locations due to increased critical minerals and metals used for the higher share of renewable energy systems within the electricity mix. In the future, Europe is the most beneficial location for *Metal use*, with the US becoming the least, despite having the lowest impact considering current mixes. This suggests the US will experience the greatest increase in renewables' share in the mix. Unlike *Climate change*, which differs considerably between locations, *Metal use* has a much smaller range. Greater changes to *Climate change* are caused by higher contributions from the wafer (and consequent high electricity contribution), this contribution is much less significant for *Metal use*, causing the smaller range. Expected silver consumption for this deployment is ~0.1 Mt. Investigations into material (e.g. silver) demand, considering different scenarios, have been compared to global production elsewhere[7], these

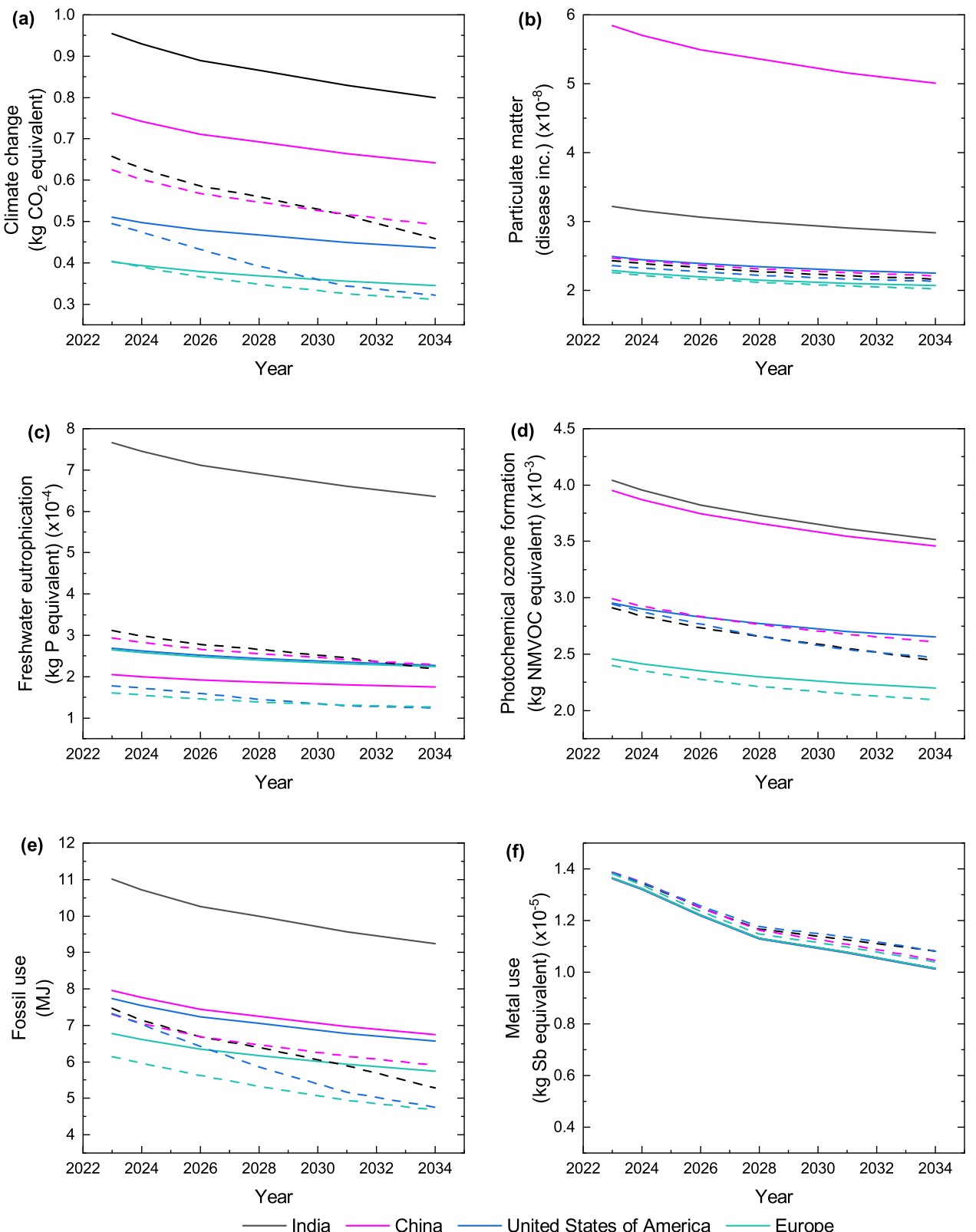

**Fig. 3 | Environmental impact of 1 W_p TOPCon manufacturing in different geographical locations.** The impact, per W_p of TOPCon module manufactured, is shown for **a** Climate change, **b** Particulate matter, **c** Freshwater eutrophication, **d** Photochemical ozone formation, **e** Fossil use and **f** Metal use. Investigated geographical locations include: India, China, the United States of America and Europe between 2023 and 2034, assuming current (solid) and future (dashed) electricity mixes, performance improvements and material developments. All modules are assumed to be transported from the location of manufacturing to central Europe.

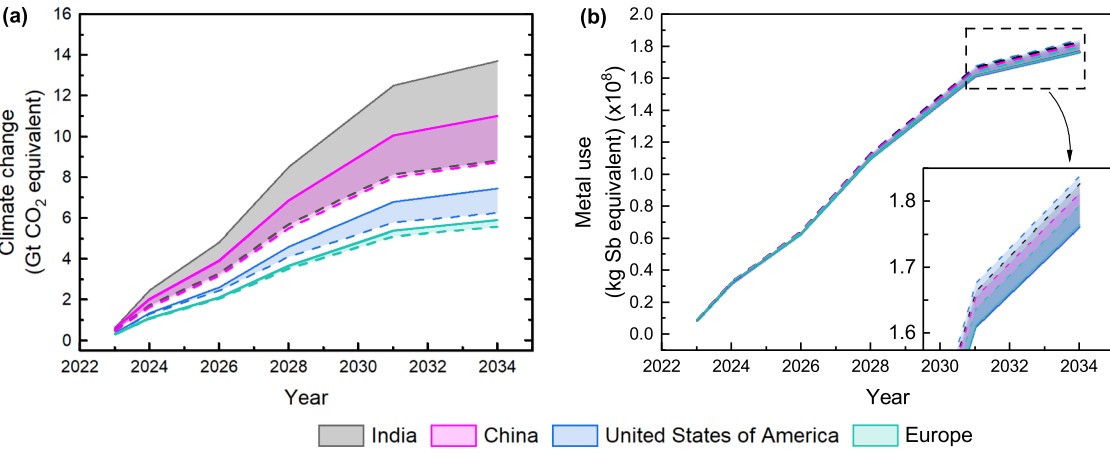

**Fig. 4 | Cumulative environmental impact from photovoltaic manufacturing due to global deployment between 2023 and 2034.** The projected global deployment of PERC and TOPCon modules is considered for the impact categories; **a** Climate change, and **b** Metal use. Current electricity mixes (solid lines) and future electricity mixes (dashed lines) are shown. The shaded regions show the range of potential impact between current and future electricity mixes for each location.

neglect changes to multiple environmental impacts, achievable using LCA.

## Carbon dioxide emissions, per kilowatt·hour

The $CO_2$ equivalent emissions are investigated for PERC and TOPCon deployment up to 2035 and compared to future electricity mixes for each location[23]. This comparison, in Fig. 5, shows that deployment of PERC and TOPCon (Solar PV) results in 0.017 kg $CO_2$ eq. kWh$^{-1}$ by 2035, whereas the carbon intensity of electricity mixes for the investigated locations is substantially higher (e.g., China and the US will emit 0.608 kg and 0.210 kg $CO_2$ eq. kWh$^{-1}$, respectively).

The PV deployed in this study will contribute 94,602 TWh of electricity generated between 2023 and 2035. This assumes an isolation of 1000 kWh m$^{-2}$ yr$^{-1}$ and a performance ratio 0.8[27], whilst considering the degradation of the PV module over time in line with ITRPV predictions. As a reminder, the assumptions are displayed in Supplementary Table 1. Carbon emissions related to this electricity generated from PERC and TOPCon are 2.26 Gt $CO_2$ eq. Generating this amount of electricity, considering future mixes, would result in 62.10, 66.75, 32.69, and 27.56 Gt $CO_2$ eq., from China, India, the US and Europe, respectively. Comparing this to the value from Solar PV shows >25 Gt $CO_2$ eq. avoided emissions. This period considers less than half the expected lifetime of PV modules (12 of 30 years), meaning the avoided emissions from this PV deployment are even greater than suggested here, providing strong incentives for mass PV deployment over the coming decade by demonstrating significant reductions in net $CO_2$. Supplementary Note 1 describes the calculations for this comparison.

## Sensitivity and uncertainty analysis

Electricity consumption is shown as a hotspot for PERC and TOPCon manufacturing. The work so far uses a single, average electricity mix for each investigated location as taken from the Ecoinvent database. However, the electricity grid within each location can differ and these sub-grids are also available in Ecoinvent and provided in Supplementary Table 6. A sensitivity analysis is conducted to consider this variation for different sub-grid electricity mixes and their effect on the *Climate change* impact of manufacturing 1 W$_p$ PERC and TOPCon. Variation is measured by using the highest and lowest carbon intensity sub-grids for each location. Results are shown in Fig. 6. The representative value of the electricity mix for each location, as used throughout the work, is shown by the black line. This analysis shows that the *Climate change* impact of PV manufacturing can vary significantly depending on which sub-grid electricity mix is used. The variation ranges between 0.32 and 0.58 kg $CO_2$ eq. W$_p^{-1}$, with the

largest variation observed for China. There is an overlap of all investigated locations and, more interestingly, it is shown that PV manufacturing in the lowest carbon intensity sub-grid location of China results in comparable $CO_2$ eq. emissions to manufacturing in (Reference) Europe. This not only demonstrates the variability of impact within each region but also the high variability in the results due to modelling choices. Another useful observation is that the same conclusions are drawn when comparing the lowest carbon intensity sub-grids to the reference electricity mix.

The sensitivity analysis is developed to investigate how the contribution of individual electricity sources to the overall electricity mix affects the impact of manufacturing a 1 W$_p$ TOPCon, for all 16 impact categories. The main contributing source in the electricity mix is varied by 5% and is compared to a Reference scenario where each of the contributing electricity sources is assumed to contribute equally. Results are summarised using relative percentage changes in Fig. 7. The raw data are found in Supplementary Table 7.

Results show that the environmental impact can change by up to 35.9% (not shown on the radar) for *Ionising radiation* when a nuclear-dominant mix is used, though typically the categories are affected by -5%. Coal has the highest impact on TOPCon manufacturing for nine categories and increasing the proportion of coal in the mix increases the impact in 12 categories, notably by +4.8% for *Climate change*. Excluding *Ionising radiation*, the impact of TOPCon manufacturing is found to be the most sensitive to coal, with an average absolute change of 2.1% and the least sensitive to PV, hydropower and biogas, all with an absolute average change 0.9%. This analysis suggests that the impact of TOPCon manufacturing is sensitive to electricity mix compositions: a relative change of 5% in source contribution can result in average absolute changes >1% for 1 W$_p$ TOPCon manufacturing for 10 impact categories. This is due to the high electricity consumption during wafer fabrication. Increasing the share of renewable energy resources in the mix can reduce the impact in most categories, as demonstrated in Fig. 7b, showing that increasing the share of any renewable electricity source decreases the impact for nine categories compared to the Reference scenario. Increased hydropower is the only scenario that experiences a decrease for all 16 impact categories, implying that hydro storage may be the most suitable, low-carbon partner to complement solar PV[28]. This sensitivity analysis is useful for providing guidance to reduce specific impact categories whilst identifying potential burden shifting.

Sensitivity analysis was also conducted on the inventory of PERC and TOPCon modules to quantify the change in environmental impact when considering technological improvement for: efficiency

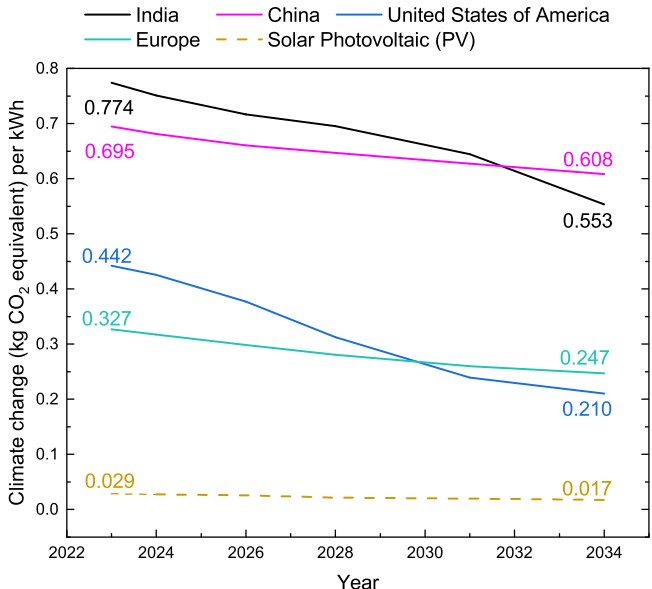

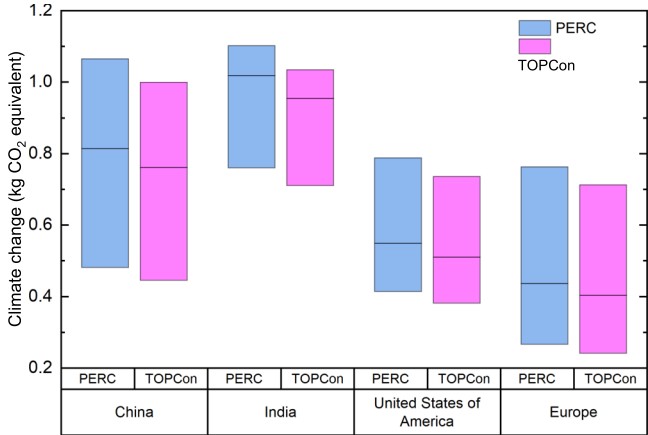

**Fig. 6 | Sensitivity analysis on *Climate change* due to sub-grid electricity mix on the impact of manufacturing 1 $W_p$ PERC and TOPCon module.** The highest and lowest carbon-intensive electricity mixes set the range of results and the reference electricity mix for each location is shown by the solid line.

**Fig. 5 | Climate change emissions (kg $CO_2$ equivalent) per kWh between 2023 and 2034.** The carbon dioxide equivalent emissions per kWh of electricity generated are investigated for the future photovoltaics (PV) deployment of PERC and TOPCon, labelled (Solar PV), and the Energy Information Administration electricity mixes of the investigated regions.

improvement, silver use reduction, wafer electricity reduction and silane usage reduction. The magnitude of the improvement and benchmark is given in Supplementary Table 8 and the raw results are available in Supplementary Table 9. The percentage change for the six highest environmental impact categories is presented in Fig. 8. Information on the modelling is outlined in the methodology. Briefly, to aid in understanding these results: the efficiency is modified to assume the module efficiency equals the stabilised cell efficiency in 2034 (Fig. 8a); silver use is assumed as 5 mg $W^{-1}$ for both technologies (Fig. 8b); electricity consumption during wafer fabrication is assumed to reduce proportionally to expected wafer thickness reductions in 2034 (Fig. 8c); and silane use is assumed to reduce (Fig. 8d) by 14.4% based on enhanced deposition rate via inductively coupled plasma-PECVD (ICP-PECVD)[29].

The percentage change due to module efficiency improvements, Fig. 8a, shows identical values for each environmental impact category. This is because the area per $W_p$, for the functional unit, is inversely proportional to the efficiency– thus the increase in efficiency will proportionally reduce the value of all impacts. Further, the scope of the LCA is limited to the manufacturing of modules, meaning there is no impact considered during the use phase, which would affect this correlation. The difference in TOPCon impact due to the efficiency improvement is 2.5% larger than the change in impact for PERC caused by the larger, relative improvement in TOPCon module efficiency (+15.9%) than for PERC efficiency (+12.6%). This suggests that the impact is sensitive to module efficiency changes and that improving module efficiency is very effective for the simultaneous reduction of PV manufacturing environmental impact across multiple categories. Reducing silver usage, Fig. 8b, is shown to be most effective for reducing the *Metal use* impact category (<41.3%) and least effective for the *Particulate matter* impact (<0.4%). Changing silver use to 5 mg $W^{-1}$ has a greater percentage reduction in silver quantity for TOPCon (78.0%) than PERC (66.5%), causing larger reductions to TOPCon categories in this figure.

Only *Metal use and Eutrophication, freshwater* exhibit impact changes >1% suggesting silver use reduction is not effective for simultaneous impact reduction across categories and that most impact categories are not very sensitive to variations in silver usage.

Wafer electricity consumption reductions are assumed equal for PERC and TOPCon (26%) resulting in impact reductions >10% in four of the six presented impact categories, shown in Fig. 8c. From these six impact categories, *Metal use* shows the smallest reductions of 0.6-0.7% for TOPCon and PERC, respectively whilst the other five presented impact categories show changes >9.6%. This is due to the higher contribution of the wafer towards these five categories and the dominance of the impact from electricity consumption, as shown by the results in Figs. 1 and 2a. The difference in *Metal use* is smaller for TOPCon than PERC, but the other five impact categories demonstrate larger impact reductions for TOPCon than PERC. This is due to the lower impact of TOPCon relative to PERC in all these impact categories, except *Metal use*, where TOPCon is higher, making the differences from wafer electricity reduction more significant in PERC manufacturing than in TOPCon. Sensitivity to silane reduction was investigated due to its identification as a hotspot during cell fabrication and its common use in silicon cell fabrication. A 14.4% reduction in both PERC and TOPCon silane usage causes reductions >0.3% for the six presented impact categories, Fig. 8d. This change is minor because the cell fabrication's contribution towards the module manufacturing is small compared to the wafer and module components. Although these changes are small, suggesting the impact is not sensitive to silane reduction, it is also shown that most of the impact categories – excluding *Metal use* – experience similar magnitudes of impact reduction (within 0.2%), demonstrating simultaneous impact category reductions.

Lastly, Monte Carlo uncertainty analysis was conducted. Results are shown in Fig. 9 and all Monte Carlo output data are accessible in Supplementary Table 10. The probability that manufacturing 1 $W_p$ PERC has a larger impact than TOPCon is >70% in most impact categories (11/16), suggesting a high level of confidence in the results. This includes *Climate change* (71.5%), *human toxicity, cancer* (85.0%), *ozone depletion* (98.7%) and *Eutrophication, freshwater* (70.21%). In contrast, *Metal use* shows 95.8% confidence that TOPCon has a higher impact than PERC. The lowest level of confidence is for *Water use*, which shows a probability of 59.3% that the *Water use* impact of PERC is larger than the impact of TOPCon. These results show that the uncertainty associated with the inventory for PERC and TOPCon does not significantly affect the overall comparison or the conclusions drawn from it.

## Discussion
Manufacturing TOPCon modules has a lower impact than PERC in 15 of 16 environmental impact categories, including a 6.5% reduction in

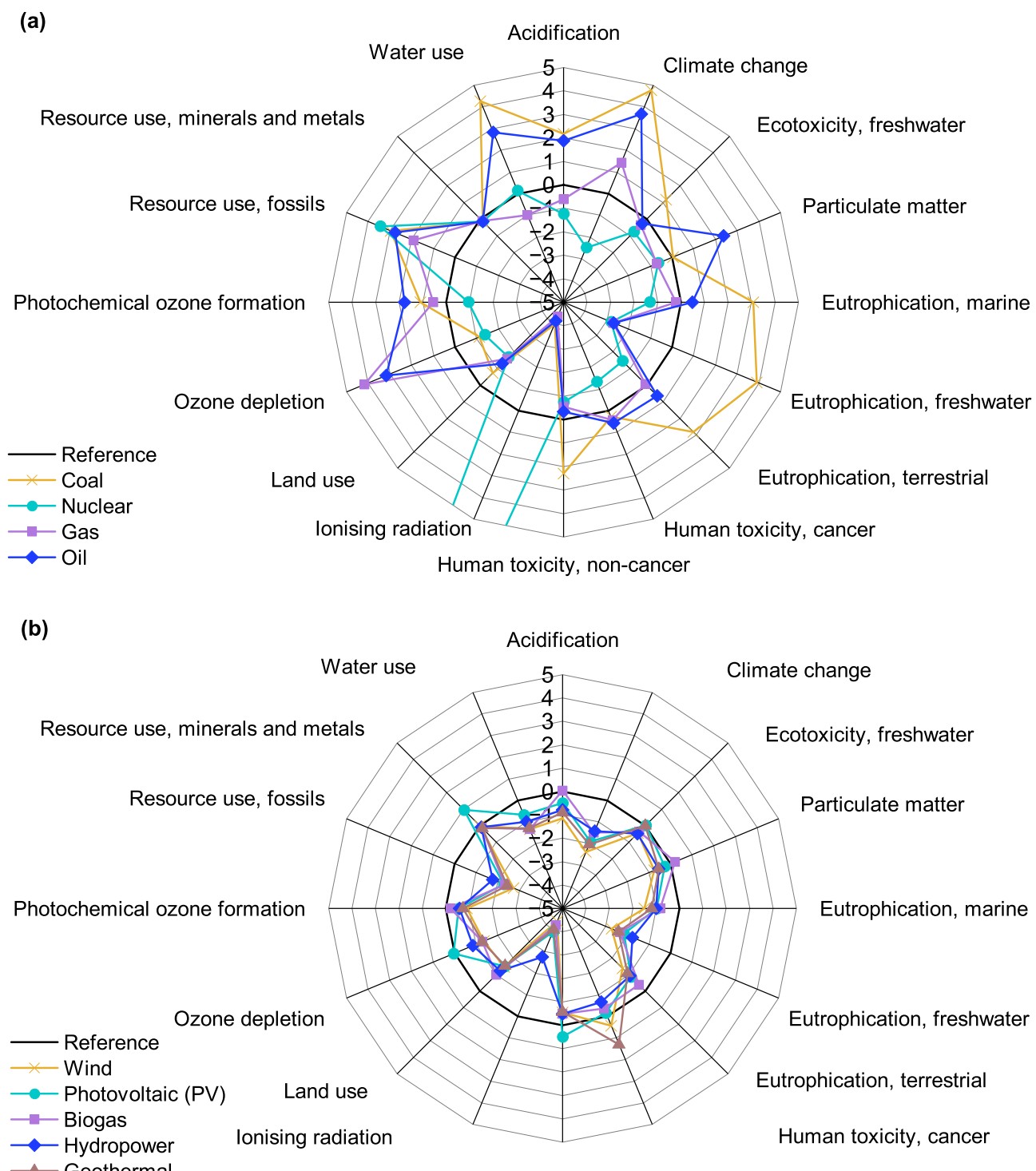

**Fig. 7 | Sensitivity analysis of the electricity mix composition on 1 W_p TOPCon manufacturing.** Radar chart showing the relative percentage change to manufacturing 1 W_p of TOPCon on each of the 16 investigated impact categories from varying the contribution to the electricity mix from each source by 5%. The radar chart is split into **a** Traditional fuel sources and **b** Renewable resources. This considers manufacturing in China using an average electricity grid mix and transportation to central Europe.

*Climate change,* per W_p. The only increased impact for TOPCon modules is *Metal use,* due to additional silver usage. Normalised results identify high value impact categories as: *Climate change, Particulate Matter, Eutrophication (freshwater), Photochemical ozone formation, Fossil use* and *Metal use.* Identified hotspots for each module manufacturing stage are: electricity consumed during wafer production, silver consumption, silane usage, copper and solar glass. Electricity consumption contributes up to 61.8% of the overall module impact in certain impact categories and is investigated further through varying manufacturing locations. Results show that, for 2023, manufacturing TOPCon modules in India has the highest environmental impact per W_p, but by 2035, China will become the highest impact location. TOPCon manufacturing location with the lowest impact to 2035 is consistently within Europe, although the US is expected to have a

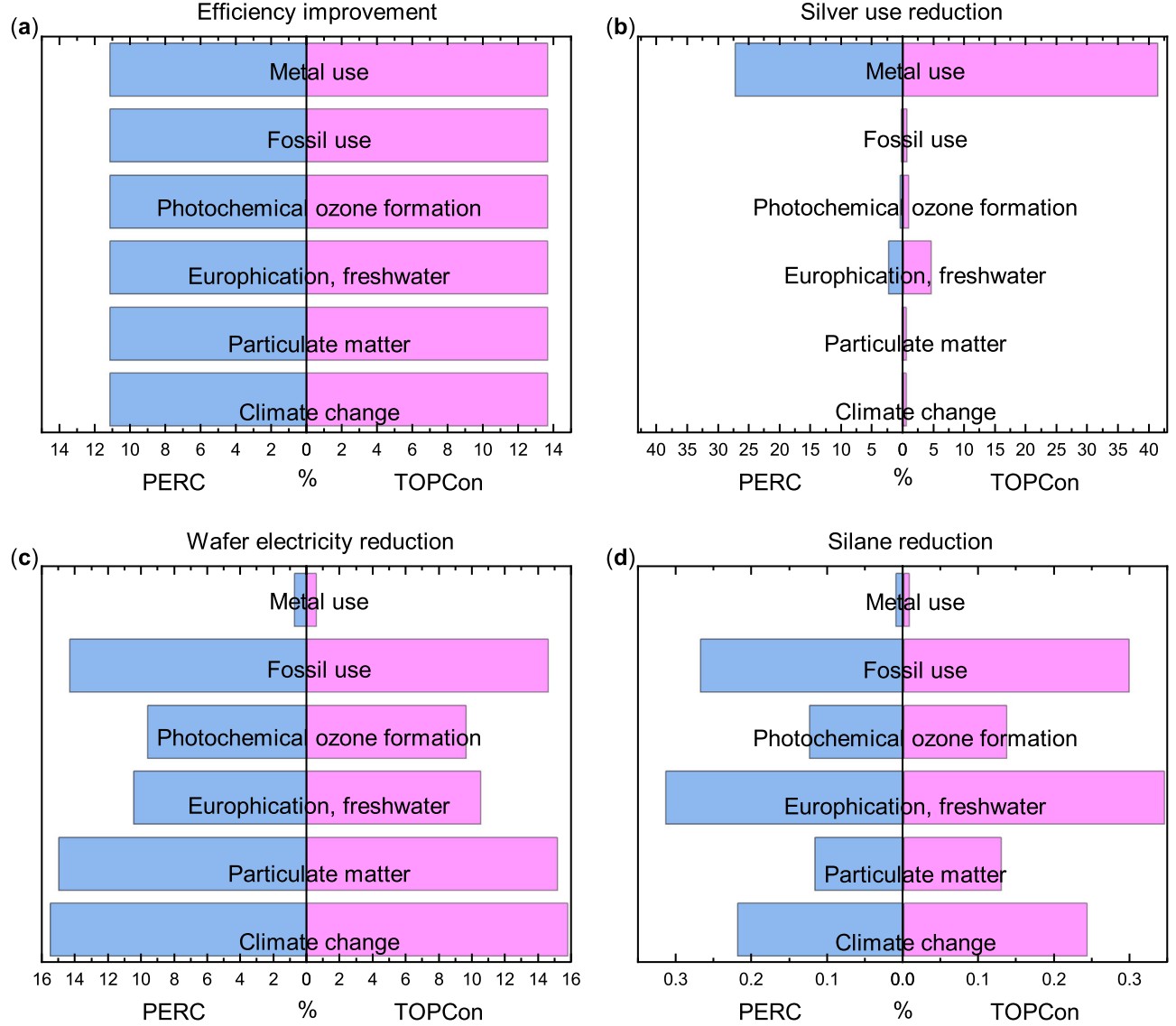

**Fig. 8 | Sensitivity Analysis showing the percentage decrease for PERC and TOPCon as a result of inventory changes compared to the baseline 2023 model, considering manufacturing in China (average mix) and transported to central Europe.** This is shown for **a** relative efficiency improvement (12.6% PERC, 15.9% TOPCon), **b** silver use reduction (66.5% PERC, 78.0% TOPCon), **c** wafer electricity reduction (26.0% for both PERC and TOPCon), and **d** silane reduction (14.4% for both PERC and TOPCon).

difference of just 3.6% *Climate change* impact per W$_P$ by 2035, considering future electricity mixes. Impact projection of global PV deployment to 2035 shows that *climate change* emissions can be significantly reduced (by 8.2 Gt CO$_2$ eq.) through manufacturing location changes. This projected deployment will result in ~500,000 TWh output from these modules throughout their lifetime. By 2035, this deployment is projected to show a total minimum reduction of 25 Gt CO$_2$ eq.

Our results for PERC are, overall, comparable to Muller et al.[11], though values differ slightly due to methodological decisions such as inventory and impact assessment method versions and different system boundaries. The difference between PERC and TOPCon also resembles those presented by Wang et al.[16]. Although their values are lower than calculated for our baseline model, the sensitivity in Fig. 6 shows overlapping results when a sub-grid mix in China is considered. This comparison is useful for seeing the variation in results for the same technology and seeing how they relate to various carbon standards. For example, for EPEAT (Electronic Product Environmental Assessment Tool) registration, PV modules meet threshold values for

EPEAT (630 kg CO$_2$ eq. kW$_P^{-1}$) and EPEAT Climate+ (400 kg CO$_2$ eq. kW$_P^{-1}$) designation[30]. Comparing the results to these values shows that PERC and TOPCon can be EPEAT registered as of 2023 – depending on manufacturing location and sub-gid mixes, as shown in Fig. 6, which includes China, the US and Europe. When current, average electricity mixes are considered, the CO$_2$ eq. values can reach <400 kg CO$_2$ eq kW$_P^{-1}$ in 2024, if manufactured in Europe with technological developments. The US is also expected to achieve this when considering future projected decarbonisation of its electricity mix, in addition to technological developments, by 2028. This modelling has not considered recycling or changes to wafer thickness, which would both further reduce the *Climate change* impact of PV modules. These results can therefore be considered a conservative estimate.

## Methods

### Life cycle assessment modelling

The results and conclusions stated in this work have been obtained using life cycle assessment (LCA). The goal of the LCA is to quantitatively investigate the environmental impact of manufacturing dual-

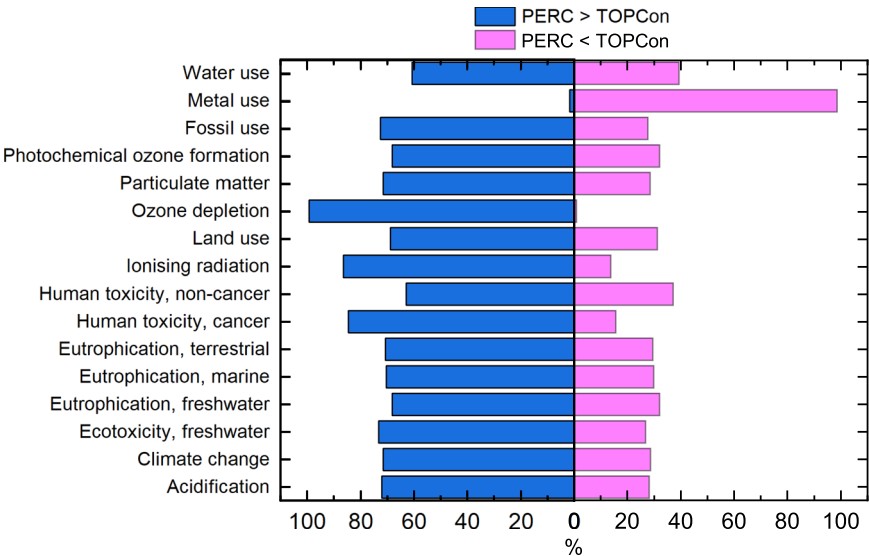

**Fig. 9 | Results from Monte Carlo uncertainty analysis with a confidence interval of 95% over 10,000 runs.** All investigated environmental impact categories are investigated to determine the confidence that the impact of manufacturing 1 $W_p$ PERC modules is greater than the impact of manufacturing 1 $W_p$ TOPCon module. This is shown by the percentage of runs where the impact of manufacturing TOPCon is less than, or otherwise greater than the impact of manufacturing PERC.

glass, TOPCon modules and assess the environmental benefits of this compared to manufacturing dual-glass PERC modules through considering the extraction of raw materials and manufacturing stages of the life cycle only, known as a "cradle-to-gate" approach. All manufacturing process stages are considered, from quartz mining to production of module components and transportation, as shown in the system boundary, Fig. 10. For all scenarios, the shipping location was fixed as central Europe. A functional unit of 1 $W_p$ was chosen to include the difference in performance of each module type. The area used to provide the functional unit is shared in Supplementary Table 11, calculated using Supplementary Equation (1) in Supplementary Data 3. Performance and material development values of PV modules are taken from the 2024 International Technology Roadmap for Photovoltaics (ITRPV)[24], which includes projections for efficiency, degradation, silver consumption and poly-Si consumption (shared in Supplementary Table 5). The foreground data used for the PERC module inventory has been taken from literature[11] and updated accordingly for current manufacturing as reflected in the ITRPV. The TOPCon module inventory is comparable to the PERC module inventory, using the same wafer and module component inventories (although changing wafer dopant to account for the n-type doping as opposed to the p-type doping used for PERC), but includes primary TOPCon cell manufacturer's data provided by an international TOPCon manufacturer. All assumptions used for the completion of this work are available in Supplementary Table 1 and the inventory for TOPCon and PERC are provided in Supplementary Table 2 and 3, respectively. The background data is provided by the Ecoinvent v3.10 database, which is used with LCA modelling software SimaPro v9.6.0.1. Environmental Footprint (EF) v3.1, as recommended by the European Union[31] is chosen to assess impact and provides a thorough analysis of 16 environmental impact categories which considers a broad spread of environmental aspects including terrestrial and aquatic ecosystems, human health, resource use and the atmosphere. This is crucial when considering a system like a photovoltaic module because it contains a wide variety of flows, including electricity consumption, heavy metals, solvents and potentially toxic chemicals, where understanding the impact of these inputs on the environment is essential for comprehensive sustainability investigations. These 16 impact categories are: *Acidification, Climate change, Ecotoxicity, (freshwater), Particulate*

*matter, Eutrophication (marine), Eutrophication (freshwater), Eutrophication (terrestrial), Human toxicity (cancer), Human toxicity (non-cancer), Ionising radiation, Land use, Ozone depletion, Photochemical ozone formation, Resource use (fossils), Resource use (minerals and metals)* and *Water use*. The results are calculated within the SimaPro modelling software to provide numerical data for midpoint impact categories, which involves using emission information from the inventory and characterisation factors from the EF v3.1 impact assessment method to calculate the LCA results. The results have also been normalised to the average annual environmental impact of a European person using assumptions from the EF v3.1 impact assessment methodology. Normalising the data allows for the identification of the high impact categories, which have been shown in the figures throughout the study. More information on the EF v3.1 impact assessment methodology can be found at[32] and a glossary for each of the impact categories can be found in Supplementary Table 2.

## Technological developments and manufacturing location considerations

Investigating alternative global regions requires changes to the electricity mix and transport inputs. The transport is considered as transporting the constructed module from the manufacturing location to a common destination in central Europe. The transport for China and Europe is taken from the same literature as the PERC inventory for consistency[11] and US transportation is taken from Grant and Hicks[33]. For India, we assume that modules are manufactured in Mundra before being transported via lorry to Gujarat[34] for shipping to a European port (Rotterdam) before being transported by lorry over a distance equal to transport inventories for other locations[11]. The electricity mixes are initially taken from the Ecoinvent V3.10 database from medium voltage, market inputs for China, the US, India and Europe to provide the baseline scenario, which does not consider decarbonisation. These inventories were last updated in December 2023 (except India, updated in March 2017), so are assumed to be accurate for the current electricity mix. Decarbonisation of each location's electricity mix is modelled on electricity generation projections taken from the most recent (2023) International Energy Outlooks'[23] "Low zero-carbon technology costs" case, which assumes a 40% reduced capital cost of

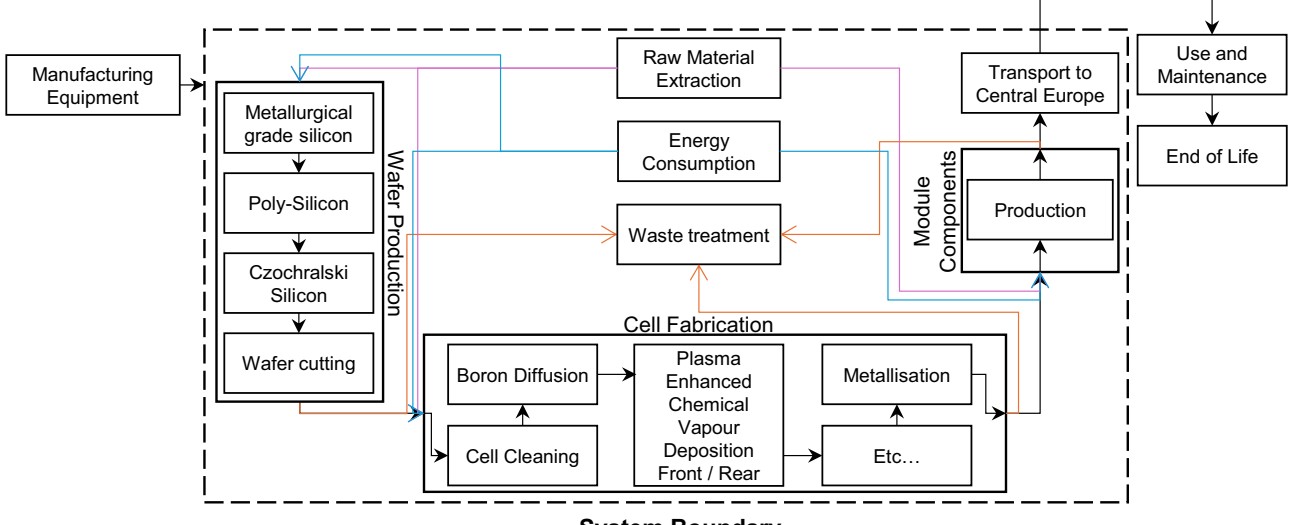

**System Boundary**

**Fig. 10 | System boundary of the photovoltaic module, TOPCon, is shown as an example.** Cradle-to-gate scope of the life cycle assessment modelling, including the key manufacturing stages of silicon wafer production, solar cell fabrication and module components such as glass, copper and ethylene-vinyl acetate used to produce a TOPCon photovoltaic module. Manufacturing is modelled as being done in China, with transportation of the photovoltaic modules to the European market. The environmental impact associated with the production of the manufacturing equipment, the use phase of the photovoltaic modules and their end-of-life are not included in the modelling and are shown as outside of the system boundary. Background data obtained from the Ecoinvent v3.10 database provides for raw material extraction and energy consumption inventories required for the foreground data for each component of the life cycle assessment modelling. Secondary data is used for wafer production, module components and transport processes, while primary data obtained from the industry is instead used to provide the TOPCon cell fabrication. Wafer production accounts for the production of the silicon ingot, which is sliced into ultra-thin wafers and passed to the cell fabrication stage. This includes the addition of dopants, passivation layers, texturing and the addition of metal electrical contacts. Wafers are then arranged in an appropriate configuration to create a photovoltaic module and connected electrically, including a junction box with a bypass diode. All waste from each manufacturing stage is considered in the life-cycle assessment modelling. Coloured arrows are used to simply clarify the flows.

zero-carbon technologies[35]. We base the analysis on this scenario due to ongoing efforts to reduce the price of renewable electricity generating sources[3]. When considering European projections, the model for "Western Europe" has been used. These have then been modelled in SimaPro LCA software such that the contribution of each electricity generation source matches the projected generation contribution mix for each year investigated: 2023, 2024, 2026, 2028, 2031 and 2035, in line with ITRPV technological projections. For years falling between EIA projections, a linear growth was assumed within each 5-year period. The electricity mix models used for future electricity mixes are provided in Supplementary Data 4.

## Global environmental impact projections

Projections for global PV deployment have also been taken from the ITRPV. This information is provided in Supplementary Table 12. The market to 2035 has been assumed to consist of only PERC and TOPCon modules, with PERC modules being completely phased out by 2034. This assumption has been made because the majority of the market (53%) is held by PERC and TOPCon modules up to the end of the ITRPV's forecasted period[24,36]. This period is also expected to experience the highest PV deployment during the journey to net-zero by 2050[1,37] hence it is expected that the impact from global PV deployment will be predominantly from PERC and TOPCon modules. Beyond 2035, and the forecast included in this study, the impact of other silicon technologies (e.g., silicon heterojunction (SHJ), interdigitated back contact (IBC)) and tandem and perovskite-based devices should be integrated into the global deployment scenarios due to their higher expected shares within the PV market. This section also calculates the benefits of global PV deployment, as shown in Supplementary Data 4, and explained in Supplementary Note 1.

## Sensitivity and uncertainty analysis

Sensitivity analysis is conducted on the electricity mix used in the LCA modelling and on various variables considered in the inventories. First, the sensitivity of the electricity mix taken from Ecoinvent v3.10 is analysed for the *Climate change* impact category, in order to consider the variation in impact due to the choice of electricity mix. In Ecoinvent, there are sub-grid electricity mixes available for each of the investigated locations: China, India, US and Europe. The highest and lowest carbon-intensive sub-grid mixes are taken for China, India and US, whereas Poland and Switzerland are selected to represent the highest and lowest carbon-intensive sub-grid mixes, respectively, within Europe. The selected inventory items from Ecoinvent are available in Supplementary Table 6. The chosen sub-grid electricity mixes were input into the LCA modelling, to replace the "average" electricity grid mixes for each location and the LCA results were recalculated for both PERC and TOPCon for the *Climate change* impact category. The sensitivity analysis on the electricity mix used was further developed to investigate how the electricity mix composition affects the impact of 1 $W_p$ TOPCon manufacturing. This is done by creating a Reference scenario where each of the investigated contributing electricity sources: coal, nuclear, gas, oil, wind, photovoltaic, biogas, hydropower and geothermal are set to 10% contribution towards the generation of 1 kWh electricity – each representing an equal contribution. The contribution of an individual electricity source is increased by 5% (whilst the other 9 contributing resources decrease proportionally – maintaining the 1 kWh output) and the impact of 1 $W_p$ TOPCon manufacturing is measured again. This allows for a comparison of TOPCon manufacturing in each of the impact categories whilst considering changes to individual contributing electricity-generating resources within the electricity mix. This is useful when looking to

target reductions for specific impact categories during PV manufacturing, with an understanding of the potential burden shift onto another impact category. All electricity mix calculations are available in Supplementary Data 4 and 6 and the source data used to create the figures are also available in the Source Data file.

Sensitivity of the following variables has also been investigated: efficiency, silver use, electricity consumption during wafer fabrication and silane use, for the six identified highest value impact categories – as determined during normalisation. The values of each variable have been modified such to represent achievable targets demonstrating the potential percentage reduction in impact as a result of these technological developments. Values used for the sensitivity analysis are available in Supplementary Table 8. The sensitivity is conducted on the baseline model, which is the assumption that the PV module is manufactured in China, using an average grid electricity mix, and transported to central Europe in the year 2023. The values for efficiency were updated from the ITRPV 2024 from the initial stabilised module efficiency in 2023, to the values of stabilised cell efficiency in 2034, also stated in the ITRPV 2024[24]. This change increases PERC efficiency by a relative 12.6% and TOPCon efficiency by a relative 15.9%. Values for silver use were initially obtained from the ITRPV 2024. For the sensitivity analysis, it is assumed that the silver consumption is decreased to just 5 mg W$^{-1}$ for both technologies since this value has been stated as a target for achieving sustainable multi-terawatt PV production[6]. The reduction is by 66.5% for PERC and 78.0% for TOPCon technology. The electricity consumption values, per m$^2$, have been taken from the PERC inventory available in the literature[11], the sensitivity analysis used new electricity values, which, for both technologies, have been reduced by 26% to represent reductions in wafer thickness by 2034 as outlined in the ITRPV 2024[24]. The values for silane have been taken from the PERC inventory in the literature[11] for the PERC module and industry verified inventory data for the TOPCon module, both of these values are reduced by 14.4%, is a conservative value based on the findings by Yoon et al.[29], who investigated inductively coupled plasma-PECVD (ICP-PECVD), which found that the introduction of H$_2$ and reduction in the silane atmospheric fraction by 14.4%, can enhance the deposition rate compared to a standard Ar/SiH$_4$ baseline.

Finally, a Monte Carlo uncertainty analysis is conducted via the built-in Monte Carlo simulator in LCA software, SimaPro v9.6.0.1. A pedigree method is used to consider the completeness, reliability, temporal, geographical and technological quality of the foreground inventory data and calculate geometric standard deviations for each, assuming a lognormal distribution. Background inventory items used Ecoinvent v3.10's built-in uncertainty parameters. The results are obtained from 10,000 runs, to ensure consistent replicability between subsequent analysis trials, with a 95% confidence level. All Monte Carlo output results are available in the Supplementary Table 10 and the pedigree matrix values are provided alongside the TOPCon and PERC cell inventories in Supplementary Data 1 and 2, respectively. The pedigree matrix for the wafer and module components inputs has been taken from the PVPS LCI publication[38].

### Assumptions and future opportunities
This work considers the impact of deployment of PERC and TOPCon up to 2035, where, beyond this period, other technological transitions within silicon PV are expected to occur[24] which would significantly increase the uncertainty level associated with the forecasted impact, as it is still unclear which cell designs will become dominant[39]. For this reason, other technologies have been excluded from this study and beyond 2035 has not been considered. Future work should look to address the sustainability of these concepts, which may include analysis of SHJ, IBC, perovskite and tandem technologies, for example.

Another area for future development is the inclusion of multiple electricity scenarios. It has been noted in this work that the current electricity mixes, taken from the Ecoinvent v3.10, can vary in terms of

when they were last updated. India, for example, was last updated more than five years ago, whilst China, the US and Europe were updated more recently, making comparison between locations difficult. To avoid this, a 2023 point was included from the chosen EIA scenario[23] (which considers low zero-carbon technology costs) to resemble the rapidly advancing field of renewable electricity and coinciding cost reduction. This is just a single scenario, which can be improved upon, though, considering multiple scenarios from a broader range of data providers. Similarly, the rate of deployment and development of TOPCon and PERC technologies has been obtained from the ITRPV[24], which has been found to show differences between projections and actual markets[40]. For this reason, future investigations should consider multiple scenarios in a sensitivity/uncertainty style analysis. Finally, the modelling used to represent changes to manufacturing location is based upon the changing transport and electricity mix inventories. In practice, each location will also have different processing yields and manufacturing capacity, which may provide interesting insights into manufacturing practices in future research.

These results are intended to be used as a guide to highlight areas of high impact associated with TOPCon manufacturing to identify future research direction as well as providing a potential forecast of the impact associated with PV deployment over the coming decade and identifying methods of reducing this through analysis of electricity scenarios. This work also only investigates the environmental sustainability of silicon PV, to provide a more rounded investigation into the sustainability of upcoming silicon PV, all three pillars of sustainability should be addressed: Environmental, Economic and Social.

## Data availability
The life cycle assessment data generated in this study are provided in the Supplementary Information and Supplementary Data files. All source data used to create figures are also provided in the Source Data file. Source data are provided with this paper.

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

## Acknowledgements

The authors gratefully acknowledge financial support for this research from the Engineering and Physical Sciences Research Council in the UK through grants EP/S023836/1 (B.L.W. EPSRC Centre for Doctoral Training in Renewable Energy Northeast Universities) and EP/W010062/1 (O.M.R., N.S.B, EPSRC Reimagining Photovoltaics Manufacturing). The work was also supported by the EPSRC Charged Oxide Inversion Layer (COIL) solar cells project (EP/V037749/1 and EP/V038605/1). S.L.P. is supported by a Royal Academy of Engineering Research Fellowship RF-2324-123-197. The authors gratefully acknowledge useful discussions with Prof Caroline Sablayrolles and Dr Claire Vialle in relation to life cycle assessment, as well as Kyle Affleck for their valuable contribution towards uncertainty and sensitivity analysis.

## Author contributions

B.L.W., O.M.R., and N.S.B. conceived and designed the research. B.L.W. and O.M.R. performed the life cycle assessment, analysis and interpretation under the supervision of N.S.B. S.L.P., N.E.G., J.D.M., and R.S.B. supported the analysis and steered the direction of the investigation. S.L.P., N.E.G., J.D.M. and R.S.B. provided technical insights relating to silicon photovoltaics and R.S.B. facilitated data collection relating to tunnel oxide passivating contact solar cells. B.L.W., O.M.R., N.S.B. prepared the manuscript and revised submission; B.L.W, O.M.R, S.L.P, N.E.G, J.D.M, R.S.B, and N.S.B reviewed and commented on the manuscript and the revision.

## Competing interests

The authors declare no competing interests.
