## [Transparent Peer Review file · Nature Communications]

Maximising environmental savings from silicon photovoltaics manufacturing to 2035

Corresponding Author: Professor Neil Beattie

Version 0:

Reviewer comments:

Reviewer #1

(Remarks to the Author)

This is a high-quality manuscript focusing on a really important topic of environmental impact projection of silicon PV manufacturing. The methodology was described clearly, data sources and results were provided sufficiently. However, the novelty and the research gap of this work was not presented due to lacking comparing with other published work in this area. There are also some improvements needed regarding clarifications and precision, please see the detailed comments attached.

Reviewer #2

(Remarks to the Author)

The paper presents a large volume of data that is rapidly changing, making it difficult for reviewers to verify each piece individually. It is requested to provide detailed sources and years for the data presented. Additionally, the most significant logical inconsistency in this paper lies in the fact that the price of TOPCon solar cell modules produced in China is much lower than in other parts of the world, so why would their CO₂ emissions be significantly higher than in other regions? The production efficiency in Europe and America is low, with very limited capacity, yet how can they achieve lower CO₂ emissions? Clearly, the aforementioned contradictions are illogical.

Reviewer #3

(Remarks to the Author)

The paper evaluates the environmental impact with transitioning from PERC to TOPCon modules. I think the paper is interesting and useful to ensure future technologies do not create new concerns. However, I do find the impact of the work rather limited and potentially misleading due to the 3 main reasons below:

- 1) it relies mostly on already published inventory data that have been updated for the wafer only based on ITRPV values.
- 2) Assumptions and scenarios considered are not clear. Typical LCA require a clear system boundary with information about input/output, what is included/excluded and what is compared. In this case the comparison is between PERC and TOPCon where most of the upstream process is considered the same (up to cell) and only modified for efficiency. The reader might be able to figure it out by reading at the same time the paper + supporting info + references but it certainly lacks transparency. A table summarizing assumptions for each cell would be required to make sure there is no bias in the approach. Also, that would be required if the author wants the study to be reused. For example, the final assumption assume transporting the module from China to central Europe (which I assume is where the authors are). If there was a system diagram and/or clear assumption table it would be easier to see how to modify the study to be more broadly applicable.
- 3) There is no uncertainty analysis conducted even though monte carlo analysis is mentioned. All results are deterministic. The only figures that show various scenarios are Fig 3 and 4 where the only variable that is changed is either the grid remains the same or change over time. Sensitivity analysis is only done on grid mix, not technology input. That's neither sensitivity or uncertainty analysis. A reduction of 6.5% in carbon footprint is small – looking at the impact of changing some of the input in the LCI would have been useful before claiming that TOPCon are better than PERC. This conclusion is also

only valid if GHG is more important than resource depletion.

In addition, minor comments:

- 4) Following up on comments #2 about lack of clarity about assumptions – Fig 2 ends up with a mix of transport from Europe and China while the original scope in the introduction mention comparing India, China, US and Europe. The abstract only mentions reduction of 6.5% in CO₂ / Wp – is that for this China to Europe scenario?
- 5) Fig 1 + 2 – what are the units for each indicator?
- 6) The authors said they run monte carlo analysis – how? None of the results show any range. All analysis seems to be deterministic. There is no information about this was done. I don't see any input with uncertainty in the LCI supporting info.
- 7) Fig 3f – use of axis is misleading – Fig F looks like there is a huge decrease but it's only because of axis choice.
- 8) A lot of the results are related to grid environmental impact. Europe is not a country and large country such as US, India and China all have sub regions grid in Ecoinvent. It easy to make Europe looks good and China bad when using this approach. Again - this could be answered with better sensitivity or scenarios analysis.
- 9) Comparison with other studies is very difficult because results are normalized. How does the new carbon footprint of PERC compare with the original reference paper with the ITRPV update? How does the results compare with carbon standards like EPEAT (goal is to be under 400 kg CO₂/kWp for ultra-low carbon) or EU Ecodesign directive ?

Version 1:

Reviewer comments:

Reviewer #1

(Remarks to the Author)

The authors has done a great job in addressing my comments. I am also glad to see we share the common understanding in few important aspects which have been corrected/improved in the revised manuscript, including enhanced description of novelty, added comparison with other research work, more cleared and precise explanation in numbers, assumption and so forth. I am happy to accept this improved manuscript for publication.

Reviewer #3

(Remarks to the Author)

Thank you for addressing the comments, in particular the contribution of this work is a lot more clear. I think for LCA being transparent and providing sufficient details to be reproducible is important and in that sense the paper is greatly improved. I have only 3 small comments regarding the figures:

Fig. 5 shows up to 2034 but legend say up to 2035, this is confusing

Fig 8 – it makes sense that efficiency has the same impact on all impact categories since LCA is based on manufacturing and there is no impact during the use phase – it might not be necessary to show this one but maybe mention it in the text only

Fig 9 – y axis is % change compared to reference? It's not in the figure or the caption

Response to Reviewers' Comments

We wish to thank all the Reviewers for their valuable feedback on our manuscript. This feedback is extremely useful, and we have carefully considered and incorporated all the Reviewers' comments into our revised submission.

Our work reports on the environmental impact of TOPCon PV manufacturing compared to PERC and provides an evaluation of this impact to 2035 for different manufacturing locations. To the best of our knowledge, this is the first time that an industrially validated TOPCon PV cell inventory has been submitted for publication into the public domain and will therefore have considerable impact. We also report the first insight into TOPCon manufacturing outside of China and the cumulative environmental impact of worldwide PV deployment to 2035 considering technology innovation and electricity mix decarbonisation in various locations.

We note that the Reviewers were positive about our work, with Reviewer 1 commenting that ***“This is a high-quality manuscript focusing on a really important topic of environmental impact projection of silicon PV manufacturing.”*** We were also pleased that Reviewer 1 found that ***the “methodology was described clearly, data sources and results were provided sufficiently.”*** We also highlight that Reviewer 3 indicates that ***“the paper is interesting and useful to ensure future technologies do not create new concerns”***.

In addition to these positive comments, all reviewers provided useful suggestions to improve our work. We have significantly revised the manuscript accordingly and a summary of the significant changes is as follows:

- addition of a more comprehensive literature review which sets a clearer context for the work and highlights its distinctiveness relative to other studies;
- addition of a “Sensitivity and uncertainty” section including three new figures;
- restructuring and reformatting of Supplementary Information and Supplementary Data documents with associated signposting in the manuscript to enhance clarity and navigation; and
- addition of descriptions and further information to the manuscript, Supplementary Information and Supplementary Data to enhance transparency.

We provide a point-by-point response to the Reviewer's comments in the tables that follow.

Contents

Page 2 – Response to Reviewer 1

Page 7 – Response to Reviewer 2

Page 8 – Response to Reviewer 3

Response to Reviewer 1

#	Comment	Response
1	Novelty and research gap of this work was not presented due to lacking comparing with other published work in this area.	We are grateful to the Reviewer for the guidance to present this more clearly including a comparison to relevant published work in the area. We have revised the introduction and literature review to make this distinction clearer. In brief, the key novelty we present is a unique combination of (a) internationally accepted projections for PV technology development with (b) global electricity mix scenarios. When both factors are used in LCA modelling, they reveal significant variations in environmental impact depending on where the PV modules are manufactured. This is the first time that this has been achieved for TOPCon outside of China. Furthermore, our work is the first to provide an industrially validated TOPCon PV cell inventory for other researchers and policy makers to use. This is important as TOPCon cell technology is expected to dominate solar electricity for the next decade. We have added a comparison to other published work in the introduction and highlighted the novelty of the present work more strongly. These changes can be found on: Page 1, Line 26-54; Page 1, 84-92; and Page 2, Line 1-3.
2	“Reduction of 8.2 Gt CO ₂ eq. by 2035” – indicate the significance of this compared to the global green-house gas emission or to other matrix to reflect the impact of this number.	We thank the reviewer for this suggestion. We have put this number in the context of 2019 global greenhouse gas emissions reported by the Intergovernmental Panel on Climate Change. Silicon PV deployment to 2035 can result in up to 13.8 Gt CO₂ eq. emissions, but changes to manufacturing location and grid decarbonisation can reduce this by 8.2 Gt CO₂ eq. This corresponds to an astounding 13.9% of global net anthropogenic greenhouse gas emissions in 2019 [IPCC report]. We have made this clear in the text as it provides a very strong measure of impact. The change in the manuscript can be found on Page 6, Line 21-23.
3	Literature review seems a missing part so the research gap is missing. As some work on comparing the environmental impact of PERC vs TOPCon technology has been published, please add a short session about this.	We thank the reviewer for this suggestion and agree that a more detailed insight into published work would be useful to readers. We are aware of just one previous publication detailing an LCA of TOPCon (Z. Wang et al., J. Clean. Prod., (2024), 470, 143187). Importantly, this other work is limited to manufacturing in China only. In contrast, our work is the first to explore the environmental impact of manufacturing TOPCon in China, India, the EU and the USA. Further, our LCA models uniquely incorporate the effects of: (i) projected improvements in materials consumption and technology; and (ii) decarbonising the electricity mix in these locations to demonstrate that this can result in a wider range of environmental impacts. Both of these effects are sourced from internationally accepted sources (the International

		Technology Roadmap for Photovoltaics and the US Energy Information Administration) that are freely accessible in the public domain. Another important difference is that our work is the first to provide an open-access and industrially validated TOPCon life-cycle inventory. This is significant because it will allow other researchers to build on the work and develop further valuable LCA modelling. As well as the additional literature added in response to the Reviewer’s first comment, we have also revised the manuscript text to describe the work of Wang et al. in more detail and subsequently identify the research gap that our work occupies. This can be found on Page 1, Line 75 – 83.
4	Recently, PV has been deployed at an unprecedented scale, reaching 1 TWp in 2022 and doubling within just two years”: please correct the statement to be more precise on the description. It is supposed to be “reaching over 1TWp of cumulative PV installed capacity by the end of 2023” according to reference [4]	Thank you for bringing this to our attention, this text has been corrected to be aligned with reference [4], Page 1, Line 23-24.
5	“TOPCon cells incorporate a silicon oxide tunnel layer at the rear, as well as the front, and rear silver contacts to improve passivation, resistance and carrier collection”. If it aims to explain it in details, the explanation of TOPCon cells should be more accurate and complete. Topcon cells incorporated thin silicon oxide layers on both front and rear sides to improve surface passivation, a heavily-dop poly silicon layer at rear side to reduced the contact resistance under the rear silver contacts.	We agree with the Reviewer that a more accurate and detailed explanation and comparison between TOPCon and PERC architecture is important for setting the context for this work. We have provided a more comprehensive description of TOPCon compared to PERC. The new text is in the manuscript Page 1, Line 57–73.
6	It is not clear which aspect of “sustainability of TOPCon modules” the authors are referring to, as sustainability involves 3 pillars, eg. environmental, social, and economic, please make the statement more clear.	This is an important point that the Reviewer has highlighted and we thank them for bringing it to our attention. We have added text to the introduction to clarify that this is environmental sustainability (Page 1, Line 74) and due to the importance of this point we have also added additional text to the “Assumptions and Future Opportunities” section, Page 14, Line 12–17.
7	“stresses the importance of minimising electricity use...”, I expect that the authors are	The Reviewer is correct in commenting that it is important to reduce the amount of electricity produced from fossil fuels however, we also note that reducing the amount of

	referring to reduce the “dirty” electricity powered by fossil fuels instead of the amount of electricity, please correct it if this is agreed.	electricity use – regardless of source – for example through process efficiency, will also result in lower environmental impacts. We have sought to clarify this by revising the manuscript on Page 2, Line 73–80.
8	“only” was repeated	Thanks for spotting this. The repetition has been removed, Page 2, Line 81.
9	“reducing poly-Si thickness” was highlighted as a way to reduce the environmental impact of TOPCon, however, reducing the thickness of poly-layer will have adverse impact on the cell performance, and some PV manufacturing has also demonstrated that including the poly-Si layer at front side as well can improve the TOPCon performance further. So please modify the last sentence of this paragraph to explain, maybe good to bring the point that even though reducing the thickness of poly-Si may adversely impact the Topcon cell performance, trade-off between cell performance and its environmental impact should be carefully considered (just an example of expression what I meant)	This is another very important point that the Reviewer has highlighted and one that we agree should be addressed. We have added text at the end of this section to highlight the trade-off between performance and environmental impact, Page 2, Line 111-116.
10	Section of “global manufacturing to 2035”, it says “We investigate the impact of TOPCon manufacturing for India, China, the US and Europe by changing the electricity mix to represent each location and modifying the transport inputs accordingly.”, but the actual results in Fig. 3 were obtained only considering the percentage of solar over the overall electricity generations if I am not wrong? If so, please clarify this in the texts to avoid misleading, particularly as “electricity mix” was highlighted in many sections of the paper. The reason is that when considering electricity mix where other factors should be included, eg.	We thank the reviewer for highlighting this confusion in our presentation of the projected impacts of TOPCon manufacturing and bringing it to our attention. To help clarify we have renamed the section “TOPCon technology development and manufacturing location”, where here we present LCA results identifying the impact of manufacturing 1 W_p of TOPCon PV in different geographical locations and accounting for the forecasted improvements in cell technology. The modelling of locations involves changing the transportation inputs (from the manufacturing location to central Europe) and the electricity mix (the composition of the electricity generating resources used for solar panel production). Electricity mixes are taken from the LCA Ecoinvent v3.10 database (representing “Current” electricity mixes) and projections to 2035 use the US Energy Information Administration (representing “Future” electricity mixes). The details of this methodology are clarified in the Supplementary Information, Supplementary Note 1.

	other metals like steel and copper in addition to silver should also be considered for wind energy.	All materials used for the various electricity generating resources – such as wind, coal and gas – have been considered through this LCA approach. We have added text to the beginning of this section to describe the results and provide a smoother introduction making sure this explanation is well communicated, Page 3, Line 39–46.
11	Figure 4 (b), the difference is difficult to be seen with the current scale, it might be good to include a magnified version to show the difference clearer.	We thank the reviewer for this suggestion, we have modified the figure to include an inset which magnifies the difference more clearly towards 2034 whilst still presenting the range of values over time. The new Figure 4 is shown below and added to the manuscript, Page 6.  Figure 4 consists of two line graphs, (a) and (b). Graph (a) shows 'Climate change (Gt CO₂ eq.)' on the y-axis (0 to 16) against 'Year' on the x-axis (2022 to 2034). It features four shaded regions representing India (grey), China (red), United States of America (blue), and Europe (green). All regions show an upward trend, with India having the highest cumulative change and Europe the lowest. Graph (b) shows 'Metal use (kg Sb eq.) (x10⁶)' on the y-axis (0.0 to 2.0) against 'Year' on the x-axis (2022 to 2034). It also features four shaded regions for India, China, USA, and Europe. An inset in graph (b) magnifies the period from 2030 to 2034, showing that the lines for all regions converge and become very close to each other, making the differences difficult to distinguish at the original scale.
12	“The PV deployed in this study will contribute 94,602 TWh of electricity generated between 2023-2035.”, what is the assumptions for the PV installation and exposed electricity generation to obtain 94,602 TWh? Please add.	We thank the reviewer for this suggestion. For clarification, the following list provides details on the assumptions used to obtain 94,602 TWh electricity generation value:  • the energy output was calculated using equation 1 from A. Müller et al. (2021, Sol. Energy Mater. Sol. Cells, 230 111277); • the insolation was 1000 kWh m⁻²yr⁻¹ (representative of Northern Europe); • the lifetime, degradation (both in year 1 and annually) and efficiency were taken from the 2024 International Roadmap for Photovoltaics; • the performance ratio was assumed to be 0.8, following International Energy Agency LCA of PV methodology guidelines; and • the number of years was taken from installation to 2035. (i.e. for modules in 2023, 12 years of output are considered but for modules in 2034 just 1 year of their output is considered). When the whole lifetime of the PV is considered, the electricity output would be 499,723 TWh. However, to keep the focus of impact to 2035 this is not mentioned in the manuscript and the emissions avoided over this longer time period have not been considered (this would also involve considering changing electricity mixes beyond 2034).

		To clarify these assumptions in the manuscript, we have updated the manuscript text, Page 6, Line 59–63, and provided information on calculating this value in Calculation 3 (Supplementary Data, Dataset 3) along with a written description in the Supplementary Information, Supplementary Note 1.
13	Please add the full name of “CC”.	Thank you to the Reviewer for identifying this abbreviation, CC has been changed to Climate change . Page 10, Line 33.
14	Supplementary information 1. Table 4 Inventory table for TOPCon and PERC cell manufacturing per m ² : the silver metallisation for TOPCon (1.2e-4 kg) is 1 order of magnitude lower than that of the PERC (4.5e-3 kg)? This seems inconsistent with the results about impact of metal use discussed in the manuscript in page 7 “The only increased impact for TOPCon modules is Metal use,”. Please check and explain.	This is a very important error that the Reviewer has identified. After careful checking, we have identified that there were two typographical errors in (now) Table 2 and Table 3 in the Supplementary Information for which we apologise and are grateful to the Reviewer for identifying. Firstly, the value for TOPCon silver metallisation (1.2E-4 kg m⁻²) was not scaled for area and this should in fact be 3.62E-3 kg m⁻². Secondly, the value for PERC (4.5E-3 kg m⁻²) is from Müller et al. (2021, Sol. Energy Mater. Sol. Cells, 230 111277) and was not updated with data from the International Roadmap for Photovoltaics. This should in fact be 2.26E-3 kg m⁻². These have both been corrected in the manuscript and we wish to reassure the Reviewer that the correct values were used in the LCA modelling and therefore this typographical error does not affect the results. We have also re-checked the other values in Supplementary Tables 2 and 3 and confirm that they are now correct.
End of Reviewer 1’s Comments		

Response to Reviewer 2

#	Comment	Response
1	The paper presents a large volume of data that is rapidly changing, making it difficult for reviewers to verify each piece individually. It is requested to provide detailed sources and years for the data presented.	We appreciate that there are a lot of data involved as the Reviewer has highlighted and agree that more details are required to help with verifying the data. To improve the accessibility of the data we have added the sources and years for the data presented, as requested, and have re-structured the Supplementary Information and Supplementary Data to make navigation of this data more straightforward which includes a contents page for both documents and all data used to plot Figures.
2	Additionally, the most significant logical inconsistency in this paper lies in the fact that the price of TOPCon solar cell modules produced in China is much lower than in other parts of the world, so why would their CO₂ emissions be significantly higher than in other regions? The production efficiency in Europe and America is low, with very limited capacity, yet how can they achieve lower CO₂ emissions? Clearly, the aforementioned contradictions are illogical.	The Reviewer raises an interesting point on the relationship between cost and CO₂ emissions and is correct in stating that PV modules manufactured in China are lower economic cost. Yet, it is worth noting that the electricity grid in China is relatively carbon intensive (in the Ecoinvent database) and this is the reason for high CO₂ emissions. We emphasise that our work does not make any correlation between the cost of PV manufacturing or production efficiency on CO₂ emissions. We have added a comment to the “Assumptions and Future Opportunities” section to clarify this and highlight that future work could expand the modelling to account for these important points raised by the Reviewer. This can be found on Page 14, Line 1-6. Our work models the environmental impact of manufacturing of TOPCon PV modules in other regions, based on the electricity mixes in those locations. The modelling implicitly uses identical manufacturing processes and capacity that are independent of location. The results show that the environmental impact of PV manufacturing is sensitive to location through the carbon intensity of the electricity mixes used in manufacturing. We have added this to a new table of assumptions to the Supplementary Information (Supplementary Table 1). We also stress that our goal is not to express a preference for one manufacturing location over another but rather to show how this can result in a wide range of emissions (and related savings opportunities). We realise that the electricity mix composition in each location can vary significantly, affecting the carbon intensity of the grid. Motivated by this understanding, a new sensitivity analysis has been added to the manuscript comparing the highest and lowest carbon intensity sub-grid regions for each location. Interestingly, this analysis (Fig. 6) shows that PV manufacturing in the lowest carbon intensity sub-grid location in China results in comparable CO₂ eq. emissions to the reference manufacturing in Europe. This is added to: Page 6, Line 77-78 and Page 7; Line 5-28; and Fig 6.

End of Reviewer 2’s Comments

Response to Reviewer 3

#	Comment	Response
1	The impact of the work is rather limited and potentially misleading due to the 3 main reasons below: 1) it relies mostly on already published inventory data that have been updated for the wafer only based on ITRPV values.	We thank the Reviewer for this feedback. Whilst we have updated a previously published PERC life cycle inventory (LCI) according to International Technology Roadmap for Photovoltaics (ITRPV) values, this is not the essential part of the manuscript nor the primary focus of the work. Instead, our work brings a completely new and industrially validated TOPCon cell LCI (available in Supplementary Table 2). We think that this is valuable, particularly because we are not aware of any other published TOPCon LCI, and therefore impactful because it can be used directly by other researchers (we apologise that this wasn't clear in the original submission and have signposted the LCI and raw LCA results in the revised manuscript). With our new TOPCon LCI, it is useful to make a comparison between the LCA results of the updated PERC cell and the TOPCon cell however, the novel part of the work is that we combine both projections for evolution of these two technologies with global electricity mix scenarios into our LCA models to identify significant variation in environmental impact.
2	Assumptions and scenarios considered are not clear. Typical LCA require a clear system boundary with information about input/output, what is included/excluded and what is compared. In this case the comparison is between PERC and TOPCon where most of the upstream process is considered the same (up to cell) and only modified for efficiency. The reader might be able to figure it out by reading at the same time the paper + supporting info + references but it certainly lacks transparency. A table summarizing assumptions for each cell would be required to make sure there is no bias in the approach. Also, that would be required if the author wants the study to be reused. For example, the final assumption assume transporting the module from China to central Europe (which I assume is where the authors are). If there was a system	We agree with the Reviewer's comment and sincerely apologise for the ambiguity. This was a weakness in the original submission which we have sought to rectify with significant changes throughout the manuscript. Specifically, we have now added:  - a system boundary diagram to the Methods section of the manuscript; - a table summarising our assumptions in the Supplementary Information with appropriate signposting in the manuscript (Supplementary Table 1); and - clarification on the transport throughout the manuscript. The system boundary (new Figure 10) is shown here for the Reviewer's reference. (Page 12) 
	diagram and/or clear assumption table it would be easier to see how to modify the study to be more broadly applicable.																																							
3	There is no uncertainty analysis conducted even though monte carlo analysis is mentioned. All results are deterministic. The only figures that show various scenarios are Fig 3 and 4 where the only variable that is changed is either the grid remains the same or change over time. Sensitivity analysis is only done on grid mix, not technology input. That's neither sensitivity or uncertainty analysis. A reduction of 6.5% in carbon footprint is small – looking at the impact of changing some of the input in the LCI would have been useful before claiming that TOPCon are better than PERC. This conclusion is also only valid if GHG is more important than resource depletion.	We appreciate and thank the Reviewer for their feedback. We acknowledge the issue with sensitivity and uncertainty analysis and have addressed this by developing and re-writing the “Sensitivity and Uncertainty” section of the manuscript. Page 6, Line 76 - Page 10, Line 29. The methods are described, Page 13, Line 1-91. This is a major revision which includes the addition of three new figures (Fig. 6, Fig. 8, Fig. 9) to the manuscript which for reference, are shown here. All the data are accessible in the Supplementary Data and Supplementary Information.  Figure 6: Climate change (kg CO₂ eq.)    Region Technology Climate change (kg CO₂ eq.)     China PERC ~0.85   TOPCon ~0.75   India PERC ~0.95   TOPCon ~0.90   United States of America PERC ~0.55   TOPCon ~0.50   Europe PERC ~0.45   TOPCon ~0.40    Figure 7: Sensitivity analysis (%)    Scenario PERC (%) TOPCon (%)     (a) Efficiency improvement ~10 ~10   (b) Silver use reduction ~35 ~35   (c) Wafer electricity reduction ~10 ~10   (d) Silane reduction ~0.2 ~0.2   	Region	Technology	Climate change (kg CO ₂ eq.)	China	PERC	~0.85	TOPCon	~0.75	India	PERC	~0.95	TOPCon	~0.90	United States of America	PERC	~0.55	TOPCon	~0.50	Europe	PERC	~0.45	TOPCon	~0.40	Scenario	PERC (%)	TOPCon (%)	(a) Efficiency improvement	~10	~10	(b) Silver use reduction	~35	~35	(c) Wafer electricity reduction	~10	~10	(d) Silane reduction	~0.2	~0.2
Region	Technology	Climate change (kg CO ₂ eq.)																																						
China	PERC	~0.85																																						
	TOPCon	~0.75																																						
India	PERC	~0.95																																						
	TOPCon	~0.90																																						
United States of America	PERC	~0.55																																						
	TOPCon	~0.50																																						
Europe	PERC	~0.45																																						
	TOPCon	~0.40																																						
Scenario	PERC (%)	TOPCon (%)																																						
(a) Efficiency improvement	~10	~10																																						
(b) Silver use reduction	~35	~35																																						
(c) Wafer electricity reduction	~10	~10																																						
(d) Silane reduction	~0.2	~0.2																																						

		 <caption>PERC vs TOPCon Comparison Data</caption>   Indicator PERC > TOPCon (Left) PERC < TOPCon (Right)    Water use6040 Resource use, minerals and metals1090 Resource use, fossils7030 Photochemical ozone formation7525 Particulate matter7030 Ozone depletion955 Land use6535 Ionising radiation8515 Human toxicity, non-cancer7030 Human toxicity, cancer8020 Eutrophication, terrestrial7525 Eutrophication, marine7030 Eutrophication, freshwater7525 Ecotoxicity, freshwater8020 Climate change7525 Acidification8020  	Indicator	PERC > TOPCon (Left)	PERC < TOPCon (Right)	Water use	60	40	Resource use, minerals and metals	10	90	Resource use, fossils	70	30	Photochemical ozone formation	75	25	Particulate matter	70	30	Ozone depletion	95	5	Land use	65	35	Ionising radiation	85	15	Human toxicity, non-cancer	70	30	Human toxicity, cancer	80	20	Eutrophication, terrestrial	75	25	Eutrophication, marine	70	30	Eutrophication, freshwater	75	25	Ecotoxicity, freshwater	80	20	Climate change	75	25	Acidification	80	20
Indicator	PERC > TOPCon (Left)	PERC < TOPCon (Right)																																																			
Water use	60	40																																																			
Resource use, minerals and metals	10	90																																																			
Resource use, fossils	70	30																																																			
Photochemical ozone formation	75	25																																																			
Particulate matter	70	30																																																			
Ozone depletion	95	5																																																			
Land use	65	35																																																			
Ionising radiation	85	15																																																			
Human toxicity, non-cancer	70	30																																																			
Human toxicity, cancer	80	20																																																			
Eutrophication, terrestrial	75	25																																																			
Eutrophication, marine	70	30																																																			
Eutrophication, freshwater	75	25																																																			
Ecotoxicity, freshwater	80	20																																																			
Climate change	75	25																																																			
Acidification	80	20																																																			
4	Following up on comments #2 about lack of clarity about assumptions – Fig 2 ends up with a mix of transport from Europe and China while the original scope in the introduction mention comparing India, China, US and Europe. The abstract only mentions reduction of 6.5% in CO₂ / Wp – is that for this China to Europe scenario?	Yes, the 6.5% reduction mentioned in the abstract is for manufacturing in China with transport to Europe and we have added text to the abstract (Page 1, Line 13) to clarify this. We have clarified the transportation in the introduction and throughout the relevant places in the Results and Methods sections. In the Results section, we initially compare PERC and TOPCon (for manufacturing in China only) and then investigate the effect of manufacturing location for TOPCon only. We have updated the first two subsection headings in the Results section (at Page 2, Line 22 and Page 3, Line 35-36) to make this distinction clearer.																																																			
5	Fig 1 + 2 – what are the units for each indicator?	Thank you for this question. The units in Figures 1 and 2 are arbitrary units (now stated in the axis title) because the results have been normalised to make the data dimensionless and allow for easier comparison across the set of impact categories. We have added an explanation of the normalisation process to the manuscript (Page 2, Line 41–47) before the presentation of the results and note that the raw data together with the individual units are available in Supplementary Table 5. The data used to create all the figures are given in Supplementary Data. Normalised results are only used for the presentation of the results within the PERC vs TOPCon comparison and the hotspot analysis (Fig. 1 and Fig 2).																																																			
6	The authors said they run monte carlo analysis – how? None of the results show any range. All analysis seems to be deterministic. There is no information about this was done. I don't see any input with uncertainty in the LCI supporting info.	We strongly value this feedback from the Reviewer since transparency is a feature which we cherish and aim to achieve when conducting an LCA. We agree that there was insufficient information to support the Monte Carlo analysis in the original manuscript and apologise for this. We have looked to improve the transparency of the work by:  • adding pedigree matrix values to the life cycle inventory (PERC and TOPCon Cells in Supplementary Tables 2 and 3), 																																																			

		 • sharing the raw data from the LCA software over 10,000 runs (Supplementary Table 9), • moving the presentation of the Monte Carlo results from the Supplementary Information into the main manuscript (Fig. 9 - Shown in Response 3) • providing a more in-depth discussion of the results. The additional text discussing the Monte Carlo results is in the sensitivity and uncertainty analysis section, Page 10, Line 14-29. The method is described in the methodology, Page 13, Line 76-91.
7	Fig 3f – use of axis is misleading – Fig F looks like there is a huge decrease but it’s only because of axis choice.	We thank the Reviewer for raising this issue. We have changed the axis on Fig. 3f to reflect that this is indeed a small decrease. The new Figure is shown here for the Reviewer’s reference. 8	A lot of the results are related to grid environmental impact. Europe is not a country and large country such as US, India and China all have sub regions grid in Ecoinvent. It easy to make Europe looks good and China bad when using this approach. Again - this could be answered with better sensitivity or scenarios analysis.	This is an important point raised by the Reviewer. We agree that the differences in impact for various sub-regions can affect the results and we have addressed this by conducting additional sensitivity analysis which compares the highest and lowest carbon intensive sub-grid mixes for each of the investigated locations. For Europe, we used Switzerland and Poland for low and high intensity respectively. As the Reviewer notes, a comparison between Europe and China is more nuanced and we have described this and the new figure in the revision to the Sensitivity and Uncertainty

		section, Page 6, Line 77 – Page 7, line 28. The additional Fig. 6 is shown in Response 3.
9	Comparison with other studies is very difficult because results are normalized. How does the new carbon footprint of PERC compare with the original reference paper with the ITRPV update? How does the results compare with carbon standards like EPEAT (goal is to be under 400 kg CO ₂ /kWp for ultra-low carbon) or EU Ecodesign directive?	We understand the importance of this point raised by the Reviewer and have looked to appropriately address this comment. To allow for readers to compare our work with others we have provided the raw results in the Supplementary Information (with appropriate signposting in the manuscript) and the data used to plot the figures in the Supplementary Data. We have also added a comparison of our results to the original reference paper (Muller et al., 2021) and the EPEAT carbon standards as suggested by the Reviewer. This has been added to the “Discussion” section of the manuscript, Page 11, Line 20-46.
End of Reviewer 3’s Comments		

Response to Reviewers' Comments

We thank the Reviewers again for their valuable and constructive feedback throughout the review process and also for their comments on our revised manuscript.

Reviewer 1's comments

The authors has done a great job in addressing my comments. I am also glad to see we share the common understanding in few important aspects which have been corrected/improved in the revised manuscript, including enhanced description of novelty, added comparison with other research work, more cleared and precise explanation in numbers, assumption and so forth. I am happy to accept this improved manuscript for publication.

Response to Reviewer 1's comments

We are grateful for the Reviewer's feedback and positive comments.

Reviewer 3's comments

Thank you for addressing the comments, in particular the contribution of this work is a lot more clear. I think for LCA being transparent and providing sufficient details to be reproducible is important and in that sense the paper is greatly improved.

I have only 3 small comments regarding the figures:

1. Fig. 5 shows up to 2034 but legend say up to 2035, this is confusing
2. Fig 8 – it makes sense that efficiency has the same impact on all impact categories since LCA is based on manufacturing and there is no impact during the use phase – it might not be necessary to show this one but maybe mention it in the text only
3. Fig 9 – y axis is % change compared to reference? It's not in the figure or the caption

Response to Reviewer 3's comments

We appreciate the Reviewer's feedback which has made our research more transparent, reproducible and robust, adding significant value to the work.

1. Thank you for identifying this. The legend for Fig. 5 has been changed to 2034 to be consistent with the graph axis.
2. We understand the Reviewer's reasoning for removing Fig 8a showing the percentage change due to efficiency differences in the sensitivity analysis however, we believe that the figure provides a useful visual aid for a non-specialist audience to clearly reflect the statements made in the text and so we have made the decision not to remove Fig. 8a. We found the Reviewer's explanation useful and have added this to the manuscript text (Sensitivity and Uncertainty Analysis section>paragraph 5>third line), which we think will further improve the clarity of this finding to the audience. *"Further, the scope of the LCA is limited to the manufacturing of modules meaning there is no impact considered during the use phase which would affect this correlation."*
3. Thank you for spotting this. The y-axis on Fig. 9 is the percentage of Monte Carlo runs that deliver a particular output. For example, 71.49 % of 10,000 runs had a result where the Climate Change impact of manufacturing 1W_p PERC was greater than that of manufacturing 1 W_p TOPCon. We have added a "%" to the axis of Fig. 9 and included the following explanation in the caption: *"This is shown by the percentage of runs where the impact of manufacturing TOPCon is less than, or otherwise greater than the impact of manufacturing PERC."*

Detailed comments:

This is a high-quality manuscript focusing on a really important topic of environmental impact projection of silicon PV manufacturing. The methodology was described clearly, data sources and results were provided sufficiently. However, the novelty and the research gap of this work was not presented due to lacking comparing with other published work in this area. There are also some improvements needed regarding clarifications and precision, please see the detailed comments below.

Page1:

1. Abstract: “reduction of 8.2 Gt CO₂ eq. by 2035”, it will be good to indicate the significance of 8.2 Gt CO₂ eq compared to the global green-house emission or to other matrix to reflect the impact of this number.
2. First 1-2 paragraph: “literature review” seems a missing part, so that it is not clear the research gaps this paper is aiming to address comparing to other works. As some work on comparing the environmental impact of PERC vs TOPCon technology has been published, please add a short session about this.
3. Paragraph 1 “Recently, PV has been deployed at an unprecedented scale, reaching 1 TWp in 2022 and doubling within just two years”: please correct the statement to be more precise on the description. It is supposed to be “reaching over 1TWp of cumulative PV installed capacity by the end of 2023” according to reference [4].
4. Paragraph 1 “TOPCon cells incorporate a silicon oxide tunnel layer at the rear, as well as the front, and rear silver contacts to improve passivation, resistance and carrier collection”. If it aims to explain it in details, the explanation of TOPCon cells should be more accurate and complete. Topcon cells incorporated thin silicon oxide layers on both front and rear sides to improve surface passivation, a heavily-dop poly silicon layer at rear side to reduced the contact resistance under the rear silver contacts.
5. Paragraph 1 last sentence: it is not clear which aspect of “sustainability of TOPCon modules” the authors are referring to, as sustainability involves 3 pillars, eg. environmental, social, and economic, please make the statement more clear.
6. Last paragraph, “stresses the importance of minimising electricity use...”, I expect that the authors are referring to reduce the “dirty” electricity powered by fossil fuels instead of the amount of electricity, please correct it if this is agreed.

Page 2:

1. Paragraph 1, 2nd line, “only” was repeated.
2. Right hand side “reducing poly-Si thickness” was highlighted as a way to reduce the environmental impact of TOPCon, however, reducing the

thickness of poly-layer will have adverse impact on the cell performance, and some PV manufacturing has also demonstrated that including the poly-Si layer at front side as well can improve the TOPCon performance further. So please modify the last sentence of this paragraph to explain, maybe good to bring the point that even though reducing the thickness of poly-Si may adversely impact the Topcon cell performance, trade-off between cell performance and its environmental impact should be carefully considered (just an example of expression what I meant)

Page 3:

1. Section of “global manufacturing to 2035”, it says “We investigate the impact of TOPCon manufacturing for India, China, the US and Europe by changing the electricity mix to represent each location and modifying the transport inputs accordingly.”, but the actual results in Fig. 3 were obtained only considering the percentage of solar over the overall electricity generations if I am not wrong? If so, please clarify this in the texts to avoid misleading, particularly as “electricity mix” was highlighted in many sections of the paper. The reason is that when considering electricity mix where other factors should be included, eg. other metals like steel and copper in addition to silver should also be considered for wind energy.

Page 5:

1. Figure 4 (b), the difference is difficult to be seen with the current scale, it might be good to include a magnified version to show the difference clearer.

Page 6:

1. Line 5 “The PV deployed in this study will contribute 94,602 TWh of electricity generated between 2023-2035.”, what is the assumptions for the PV installation and exposed electricity generation to obtain 94,602 TWh? Please add.

Page 9:

1. Line 16: please add the full name of “CC”.

Supplementary information

1. Table 4 Inventory table for TOPCon and PERC cell manufacturing per m²: the silver metallisation for TOPCon (1.2e-4 kg) is 1 order of magnitude lower than that of the PERC (4.5e-3 kg)? This seems inconsistent with the results about impact of metal use discussed in the manuscript in page 7 “The only increased impact for TOPCon modules is Metal use.”. Please check and explain.